# Rejecting Hallucinated State Targets during Planning

**Mingde "Harry" Zhao** [1 2 3]  **Tristan Sylvain** [4]  **Romain Laroche** [3]  **Doina Precup** [1 2 5]  **Yoshua Bengio** [1 6]

## Abstract

In planning processes of computational decision-making agents, generative or predictive models are often used as "generators" to propose "targets" representing sets of expected or desirable states. Unfortunately, learned models inevitably hallucinate infeasible targets that can cause delusional behaviors and safety concerns. We first investigate the kinds of infeasible targets that generators can hallucinate. Then, we devise a strategy to identify and reject infeasible targets by learning a target feasibility evaluator. To ensure that the evaluator is robust and non-delusional, we adopted a design choice combining off-policy compatible learning rule, distributional architecture, and data augmentation based on hindsight relabeling. Attaching to a planning agent, the designed evaluator learns by observing the agent's interactions with the environment and the targets produced by its generator, without the need to change the agent or its generator. Our controlled experiments show significant reductions in delusional behaviors and performance improvements for various kinds of existing agents.

## 1. Introduction

The advent of computational modeling has spurred advancements in computational decision making agents, most notable of which is, model-based Reinforcement Learning (RL). This work is focused on those models that play the role of *generators*, which imagine or specify future outcomes for agents. Some generators imagine next states or observations, while others specify subgoals (sets of states) to accomplish. No matter how they are represented, we refer to these generator outputs as *targets* and methods that make such use of generators *Target-Assisted Planning (TAP)*.

TAP is a new perspective for unifying many planning / reasoning agents with wildly different behaviors. For instance,

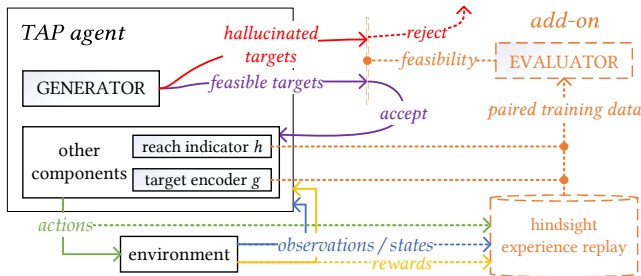

*Figure 1.* **Target Evaluator attached to a Target-Assisted Planning (TAP) Agent**: An abstracted framework that encompasses, but is not limited to, methods listed in Tab. 2 (in Appendix). The *generator* proposes candidate target embeddings $g^{\odot}$. Our proposed *evaluator* can be attached to learn to reject infeasible targets (related parts marked in dashed lines).

some rollout-based model-based RL methods are TAP, such as those following the classic Dyna or Monte-Carlo tree-search frameworks, utilize fixed-horizon transition models to simulate experiences (Sutton, 1991; Schrittwieser et al., 2019; Łukasz Kaiser et al., 2020); While, TAP also encompasses methods that directly generate arbitrarily distant targets acting as candidate sub-goals to divide-and-conquer the tasks into smaller, more manageable steps (Zadem et al., 2024; Zhao et al., 2024; Lo et al., 2024).

It is the defining characteristic of TAP - the usage of generators, that brings us to the topic of this work: an often unstated assumption is that generated targets are always feasible. However, the desired generalization abilities of generative models are inevitably accompanied by *hallucinations* (Xu et al., 2025; Zhang et al., 2024b; Xing et al., 2024; Jesson et al., 2024; Aithal et al., 2024) - the "dark side" that produces infeasible targets that can never be experienced by anyhow. Hallucinations impact TAP agents differently based on their planning behaviors. In *decision-time* TAP methods (Alver & Precup, 2024), where models are used to make an immediate decision on what to do next, hallucinated targets can lead to delusional plans that compromise performance and safety (Langosco et al., 2022; Bengio et al., 2024). For *background* TAP agents that train on simulated experiences constructed with generated targets, delusional values estimated from hallucinated targets can be catastrophically destabilizing (Jafferjee et al., 2020; Lo et al., 2024).

Human brains reject infeasible intentions hallucinated by the belief formation system (brain's generator) using a be-

[1] Mila (Quebec AI Institute) [2] McGill University [3] Wayve [4] RBC Borealis [5] Google Deepmind [6] Université de Montréal. Correspondence to: Mingde "Harry" Zhao <mingde.zhao@mail.mcgill.ca>.

*Proceedings of the 42nd International Conference on Machine Learning*, Vancouver, Canada. PMLR 267, 2025. Copyright 2025 by the author(s).

lief evaluation system (Kiran & Chaudhury, 2009). Taking inspirations, we propose to assist existing TAP agents with a target evaluator, which can be used to reject infeasible targets and in turn delusional behaviors. To maximize compatibility, the evaluator is designed as a minimally-intrusive add-on to attach to existing TAP agents Zhao et al. (2020): it learns by observing the TAP agent's interactions with the environment and the targets produced, without the need to change the agent's behaviors or the architecture. Our main contributions are as follows:

1. We systematically categorized and characterized infeasible targets *w.r.t.* time horizons

2. We discussed the desiderata of learning a minimally-intrusive non-delusional target evaluator

3. We proposed a combination of a) off-policy compatible update rule that enables the evaluator to learn by observing, b) an evaluator architecture compatible with different time horizons that enables a unified solution for most TAP agents and c) two assistive hindsight relabeling strategies performing data augmentation to provide training data beyond those collected via interactions, which ensure that the evaluator itself does not produce delusional evaluations

4. We implemented the solution, as illustrated in Fig. 1, for several types of existing TAP agents, as discussed in Tab. 2 (in Appendix), and showed that agents can better manage generated targets, reduce delusional behaviors and significantly improve performance

## 2. Preliminaries

**Problem Setting**: For the readers' ease of understanding, in this work, we use RL agents as a representative of generic computational sequential decision-making systems. We model the interaction of an RL agent with its environment as a Markov Decision Process (MDP) $\mathcal{M} \equiv \langle \mathcal{S}, \mathcal{A}, P, R, d, \gamma \rangle$, where $\mathcal{S}$ and $\mathcal{A}$ are the sets of possible states and actions, $P : \mathcal{S} \times \mathcal{A} \rightarrow \text{Dist}(\mathcal{S})$ defines the state transitions, $R : \mathcal{S} \times \mathcal{A} \times \mathcal{S} \rightarrow \mathbb{R}$ defines the rewards, $d : \mathcal{S} \rightarrow \text{Dist}(\mathcal{S})$ is the initial state distribution, and $\gamma \in (0, 1]$ is a discount factor. An agent needs to improve its policy $\pi : \mathcal{S} \rightarrow \text{Dist}(\mathcal{A})$ to maximize the value, *i.e.*, the expected discounted cumulative return $\mathbb{E}_{\pi,P}[\sum_{t=0}^{T_\perp} \gamma^t R(S_t, A_t, S_{t+1})|S_0 \sim d]$, where $T_\perp$ denotes the timestep when the episode terminates. Some environments are partially observable, which means that instead of a state, the agent receives an observation $x_{t+1}$, used to infer the state (possibly with episodic history).

**Targets**: for generality, we consider a target to be an embedding representing a set of states. Each target $\boldsymbol{g}^{\odot} \mapsto \{s^{\odot}\}$ is paired with an indicator function $h$ outputting $h(s', \boldsymbol{g}^{\odot}) = 1$ if $s' \in \{s^{\odot}\}$ and 0 otherwise. For the interest of time hori-

zons, we also introduce $\tau$ - the maximum number of time steps an agent is allowed to reach a state in $\boldsymbol{g}^{\odot}$.

Let $D_\pi(s, \boldsymbol{g})$ represents $1^{st}$ timestep $t$ *s.t.* $h(s_t, \boldsymbol{g}^{\odot}) = 1$, if $\pi$ (conditional or not) is followed from state $s$. We define $\tau$-**feasibility** of $\boldsymbol{g}^{\odot}$ from state $s$ under $\pi$ as $p(D_\pi(s, \boldsymbol{g}^{\odot}) \leq \tau) := \sum_{t=1}^{\tau} p(D_\pi(s, \boldsymbol{g}^{\odot}) = t)$. $\boldsymbol{g}^{\odot}$ is called $\tau$-**feasible** if $p(D_\pi(s, \boldsymbol{g}^{\odot}) \leq \tau) > 0$, and $\tau$-**infeasible** otherwise. Notably, $\tau$-feasibility is an intrinsic property of a state-policy-target tuple, not a heuristic measure.

A target is generally evaluated as "good" if it leads to rewarding outcomes, *i.e.*:

$$
\begin{aligned}
&\mathcal{U}_{\pi,\mu}(s, \boldsymbol{g}^{\odot}, \tau) := \\
&r_\pi(s, \boldsymbol{g}^{\odot}, \tau) + \gamma_\pi(s, \boldsymbol{g}^{\odot}, \tau) \cdot V_\mu(s_{\min(D_\pi(s, \boldsymbol{g}^{\odot}), \tau)})
\end{aligned}
\tag{1}
$$

where $\min(D_\pi(s, \boldsymbol{g}^{\odot}), \tau)$ denotes the timestep when the commitment to $\boldsymbol{g}^{\odot}$ is terminated (by $h$ or $\tau$), $s_{\min(D_\pi(s, \boldsymbol{g}^{\odot}), \tau)}$ is the state the agent ended up in, $r_\pi(s, \boldsymbol{g}^{\odot}, \tau) := \sum_{t=1}^{\min(D_\pi(s, \boldsymbol{g}^{\odot}), \tau)} \gamma^{t-1} r_t$ is the cumulative discounted reward along the way, $\gamma_\pi(s, \boldsymbol{g}^{\odot}, \tau) := \gamma^{\min(D_\pi(s, \boldsymbol{g}^{\odot}), \tau)}$ is the cumulative discount, and $V_\mu(\cdots)$ is the future value for following $\mu$ afterwards.

Eq. 1 shows that if $\boldsymbol{g}^{\odot}$ is $\tau$-infeasible, *i.e.*, $s_{min(D_\pi, \tau)} \notin \boldsymbol{g}^{\odot}$, then TAP methods blindly using $\boldsymbol{g}^{\odot}$ to determine $V_\mu$ will produce delusional evaluations - the cause of delusional planning behaviors. For example, feasibility-unaware methods, *e.g.*, Sutton (1991); Schrittwieser et al. (2019); Hafner et al. (2025), assume that targets are always reachable as long as they can be generated. Meanwhile, planned trajectories involving infeasible targets are delusional; There are also some feasibility-aware methods, *e.g.* Nasiriany et al. (2019); Zhao et al. (2024); Lo et al. (2024), in which agents estimate certain metrics to decide if a target is feasible. However, as we will discuss later, they often produce incorrect estimates, thus may still favor infeasible targets. In later sections, we propose an evaluator that simultaneously estimates the $\tau$-feasibility and $D_\pi$ of the proposed targets, where these estimations are used to decide if the evaluation of a target should be trusted or if the target should be rejected.

**Source-Target Pairs & Hindsight Relabeling** To learn the feasibility of a target from a given state, "source-target pairs" are needed, which are tuples involving a source state and a target embedding. The quality of these paired training data is critical for the training outcome (Dai et al., 2021; Moro et al., 2022; Davchev et al., 2022). Hindsight Experience Replay (HER) can be seen as a way to enhance the diversity of the pairs, by re-using targets that happened to have been achieved on existing trajectories, and pretending that they were the chosen targets during the interactions (Andrychowicz et al., 2017). HER augments a transition with an additional state $s^{\odot}$, which can be passed through a target embedding function $g$ to acquire the target embed-

*Table 1.* **Categorization of Targets based on Composition, Characteristics, Risks & Delusion Mitigation Strategies**

| Target Composition | State Correspondence | $\infty$-Feasibility $p(D_\pi(s, \boldsymbol{g}^{\odot}) < \infty)$ | Feasibility Errors & Resulting Delusional Planning Behaviors | Data Augmentation (Relabeling) Strategies against Feasibility Delusions |
|---|---|---|---|---|
| **Only or Single G.0** | non-hallucinated feasible states from $s$ | $> 0$ | **E.0**: May think G.0 states are infeasible, thus turn to riskier alternatives, *e.g.*, G.1 or G.2 | EPISODE for G.0 (and G.2) states in the same episode + PERTASK for G.0 (and G.2) beyond the episode |
| **Only or Single G.1** | hallucinated "states" not belonging to the MDP | should $= 0$ | **E.1**: May think G.1 states are favorable, thus commit to them. Impacted by ill-defined $V_\mu(\cdots)$ | GENERATE for G.1 (and G.0 & G.2) states, to be proposed by the generator |
| **Only or Single G.2** | hallucinated MDP states infeasible from $s$ | should $= 0$ | **E.2**: May think G.2 states are favorable, thus commit to them | PERTASK for G.2 (and G.0) beyond episode + EPISODE for G.2 (and G.0) states in the same episode |
| **Some G.0** | at least one non-hallucinated state from $s$ | $= p(D_\pi(s, \boldsymbol{g}^{\odot}_-) < \infty)$ $> 0$ (Thm. 4.1) | **E.0** | EPISODE + PERTASK |
| **Only G.1 & G.2** | set of ONLY hallucinated states | should $= 0$ | **E.1 & E.2** | GENERATE or GENERATE + PERTASK |

ding $g(s^{\odot})$ as the relabeled target. **Relabeling strategies**, corresponding to how $s^{\odot}$ is obtained, are critical for HER's performance (Shams & Fevens, 2022; Eysenbach et al., 2021). Most existing relabeling strategies are *trajectory-level*, meaning that $s^{\odot}$ comes from the same trajectory as $s_t$. These include FUTURE, where $s^{\odot} \leftarrow s_{t'}$ with $t' > t$, and EPISODE, with $0 \leq t' \leq T_\perp$. HER greatly enhanced the sample efficiency of learning about experienced targets. Meanwhile, its limitations to learning about only experienced targets planted a hidden risk of delusions towards hallucinated targets for TAP agents, to be discussed later.

# 3. Hallucinated Targets in Planning

Categorizing targets proposed by the generator helps inform us about how to correctly learn the feasibility of targets.

Let us warm-up with singleton targets, *i.e.*, $\boldsymbol{g}^{\odot}$ has a single element $\hat{s}^{\odot}$, and propose a characterization of singleton targets into 3 *disjoint* categories. Then, we will extend to the non-singleton targets composed of these 3 "ingredients".

## 3.1. G.0: $\infty$-Feasible

Given source state $s$, a generated singleton target $\boldsymbol{g}^{\odot}$ is called G.0 if it maps to one state which is $\infty$-feasible from $s$, with some policy $\pi$. Note that G.0 includes $\tau$-infeasible states for given finite values of $\tau$.

## 3.2. G.1 - Permanently Infeasible (Hallucinated)

G.1 includes generated "states" that do not belong to the MDP at all, *i.e.*, a target "state" $\hat{s}^{\odot}$ is G.1 if $\forall s, \pi, \ p(D_\pi(s, \hat{s}^{\odot}) < \infty) = 0$.

## 3.3. G.2 - Temporarily Infeasible (Hallucinated)

This type includes those MDP states that are *currently infeasible from state $s$*. Unlike G.1, G.2 states could be G.0

if they were evaluated from a different source state. G.2s can often be overlooked, not only because hallucinations are mostly studied in contexts that do consider the source state $s$, but also because they only exist in some special MDPs.

## 3.4. Examples

To provide intuition about these concepts, we use the MiniGrid platform to create a set of fully-observable environments, minimizing extraneous factors to focus on the targets (Chevalier-Boisvert et al., 2023). We call this environment `SwordShieldMonster` (SSM for short); In SSM, agents navigate by moving one step at a time in one of four directions across fields of randomly placed, episode-terminating lava-traps, while searching for both a sword and a shield to defeat a monster with a terminal reward. The lava-traps' density is controlled by a difficulty parameter $\delta$, but there is always a feasible path to success. Approaching the monster without both the randomly placed sword and shield ends the episode. Once acquired, either of the two items cannot be dropped, leading to a state space where not all states are accessible from the others. Thus, SSM states are partitioned into 4 semantic classes, defined by 2 indicators for sword and shield possession. For example, $\langle 0, 1 \rangle$ denotes "sword not acquired, shield acquired".

G.1 generations in this environment may be semantically valid, *e.g.*, an SSM "state" with the agent surrounded by lava, as in Fig. 2 (top row), or totally absurd, *e.g.*, an SSM observation without an agent.

G.2 states can be once G.0 but are now blocked due to a past transition, *e.g.*, after acquiring the sword in SSM, the agent transitions from class $\langle 0, 0 \rangle$ to $\langle 1, 0 \rangle$, sealing off access to $\langle 0, 0 \rangle$ or $\langle 0, 1 \rangle$; G.2 can also appear due to the initial state distribution $d$: some states can only be accessed from specific initial states, *e.g.*, an agent spawned in $\langle 1, 0 \rangle$ cannot reach $\langle 0, 0 \rangle$ or $\langle 0, 1 \rangle$. An example of delusional behavior caused by a G.2 target is provided in Fig. 2 (bottom row).

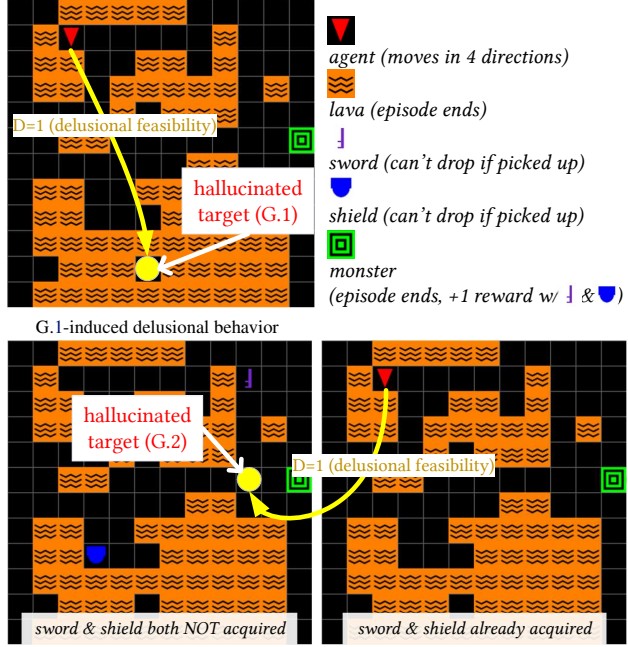

*agent (moves in 4 directions)*

*lava (episode ends)*

*sword (can't drop if picked up)*

*shield (can't drop if picked up)*

*monster*
*(episode ends, +1 reward w/ ⌡ & 🛡)*

G.1-induced delusional behavior

G.2-induced delusional behavior (*s* in $\langle 1, 1 \rangle$, $s^{\odot}$ in $\langle 0, 0 \rangle$)

*Figure 2.* **Delusional Plans in** `SSM`: In both cases, a TAP agent, lacking understanding about the hallucinated targets (yellow dots), mis-evaluates their feasibility, leading to delusional plans which suggest that the task goal can be achieved via the infeasible targets.

Despite rising concerns regarding the safety of TAP agents (Bengio et al., 2024), their delusional behaviors remain under-investigated, largely due to the **lack of access** to ground truths needed to identify hallucinations and their resulting delusional behaviors. Thus, it is critical to analyze with clear examples and conduct rigorous controlled experiments where the ground truth of targets could be solved with Dynamic Programming (DP) (Howard, 1960), which is why we created `SSM` and used it later for experiments.

### 3.5. Non-Singleton Targets

For the general case, where a generated target embedding $\boldsymbol{g}^{\odot}$ potentially corresponds to a set of "states" $\{\hat{s}^{\odot}\}$, the elements of the associated set may span all categories, *i.e.*, the target can correspond to a mixture of G.0, G.1 & G.2 states. Tab. 1 summarizes the possible target compositions and their properties[1].

# 4. Evaluating Targets Correctly and Robustly

Knowing that hallucinations cannot be eradicated in general, we intend to lower their risks by adopting the brain-inspired solution - to reject infeasible targets post-generation. If done effectively, the negative impact of hallucinated tar-

---

[1]There may be no explicit mapping from a target embedding to a set of "states" and thus any target can always map to arbitrarily many G.1 "states". Thm. 4.1 explains that this problem is benign.

gets becomes limited to the resource cost of generating and rejecting targets, to be discussed in detail. This approach is in contrast with directly trying to address hallucinations in the generators case-by-case, which we deem to have an unbreakable glass ceiling and not versatile enough to be generalized to generic TAP methods.

For a feasibility evaluator to be effectively differentiate the proposed targets, it should correctly estimate the feasibility of targets which maps to all kinds of states (G.0, G.1 & G.2). However, learning to estimate feasibility is not as trivial as it seems, because improper training could naturally lead to delusional feasibility estimations, which cannot be simply addressed by scaling up training. If the evaluator has delusions of feasibility, then its incorporation becomes futile, as hallucinated targets could still be favored.

For estimation errors, we similarly warm up with those of the singleton targets. For clarity, we use matching identifiers E.0, E.1, and E.2 to denote the estimation errors of feasibility towards G.0, G.1, and G.2 "states", respectively. These discussions are presented in Tab. 1.

When targets correspond to general sets of states, the following holds:

**Theorem 4.1.** *Let $\boldsymbol{g}^{\odot}$ be a target embedding. Its feasibility from state s satisfies:*

$$\forall \pi, p(D_{\pi}(s, \boldsymbol{g}^{\odot}) \leq \tau) = p(D_{\pi}(s, \boldsymbol{g}^{\odot}_{-}) \leq \tau)$$

*where $\boldsymbol{g}^{\odot}_{-}$ is a target that correspond to the set of states of $\boldsymbol{g}^{\odot}$ with all infeasible states (G.1 & G.2) removed.*

This result indicates that a target is infeasible if and only if it consists entirely of infeasible states, allowing us to focus on learning processes that identify such cases.

### 4.1. Desiderata for Evaluator

We used the following important considerations to guide our design for an appropriate feasibility evaluator.

- **[automatic]** the evaluator must learn to automatically differentiate the feasibility of all kinds of targets without pre-labeling: *we need to exploit h*

- **[minimally intrusive]** the evaluator should be generally applicable to existing TAP agents, without changing the agents too much to disturb the generally-sensitive RL components: *we need to ensure its behavior as an add-on and it can be conditioned on the policy π of the agent, to learn alongside the agent by merely observing*

- **[unified]** the evaluator should have a unified behavior compatible with different τs: *we can design it in a way to learn the τ-feasibilities for many τ values simultaneously*

### 4.2. Learning Rule & Architecture for Feasibility

Following the considerations, we propose to use the following learning rule to *indirectly* learn the targets' feasibility

by *directly* learning the distribution of $D_\pi(s, \boldsymbol{g}^\odot)$.

$$D_\pi(s, \boldsymbol{g}^\odot) \leftarrow 1 + D_\pi(s', \boldsymbol{g}^\odot) \text{ , with} \qquad (2)$$

$$\begin{cases} D_\pi(s, \boldsymbol{g}^\odot) \equiv D_\pi(s, a, \boldsymbol{g}^\odot), a \sim \pi(\cdot|s, \boldsymbol{g}^\odot) \\ D_\pi(s', \boldsymbol{g}^\odot) := \infty \text{ if } s' \text{is terminal and } h(s', \boldsymbol{g}^\odot) = 0 \\ D_\pi(s', \boldsymbol{g}^\odot) := 0 \text{ if } h(s', \boldsymbol{g}^\odot) = 1 \end{cases}$$

This results in an off-policy compatible policy evaluation process over a parallel MDP almost-identical to the task MDP, but adapted for $\boldsymbol{g}^\odot$, where all transitions yield "reward" $+1$ and states satisfying $\boldsymbol{g}^\odot$ are changed to terminal with state value 0. Every time an infeasible target embedding is sampled for training, the update rule will gradually push the estimate towards $\infty$, for all sampled source state $s$.

Our design only learns $D_\pi$ in a way that can lead to $\tau$-feasibilities $p(D_\pi(s, \boldsymbol{g}^\odot) \leq \tau)$. For this purpose and the consideration for a unified design, we propose to use Eq. 2 in conjunction with a C51-style distributional architecture (Bellemare et al., 2017), which outputs a distribution represented by a histogram over pre-defined supports. When we set the support of the estimated $D_\pi(s, \boldsymbol{g}^\odot)$ to be $[1, 2, \cdots, T]$ with $T$ sufficiently large, the learned histogram bins via Eq. 2 will correspond to the probabilities of $p(D_\pi(s, \boldsymbol{g}^\odot) = t)$ for all $t \in \{1, \ldots, T-1\}$. This technique of using C51 distributional learning enables the extraction of $\tau$-feasibility $p(D_\pi(s, \boldsymbol{g}^\odot) \leq \tau)$ from a learned $T$-feasibility with $p(D_\pi(s, \boldsymbol{g}^\odot) = t)$ over $t \in \{1, \ldots, T\}$, thus learning all $\tau$-feasibility with $\tau < T$ *simultaneously*. Take the example of the 1-step Dyna agent we implemented for experiments (Sec. 5.2): if the estimated histogram has little probability density for $p(D_\pi(s, \boldsymbol{g}^\odot) = 1)$, then the target (simulated next state) is likely hallucinated and should be rejected, avoiding a potential delusional value update.

Note that the C51 architecture also allows us to extract the distribution of $\gamma_\pi(s, \boldsymbol{g}^\odot, \tau)$, which, as defined in Sec. 2, the cumulative discount with a chosen target. This is done via transplanting the output histogram of $D_\pi(s, \boldsymbol{g}^\odot)$ over $[1, 2, \ldots, \tau, \tau+1, \tau+2, \ldots]$ onto the changed support of $[\gamma^1, \gamma^2, \cdots, \gamma^\tau, \gamma^\tau, \gamma^\tau, \ldots]$.

## 4.3. Training Data for Feasibility

With the proper learning rule and architecture, we now need to ensure that the evaluator have proper training data and does not become delusional. In Sec. 2, we mentioned the incompleteness of the relabeling strategies, which will be discussed in detail now: **1)** Certain relabeling strategies naturally create exposure issues, even for G.0 targets. For instance, FUTURE only relabels with future observations, thus only exposes a learner to future feasible targets, confusing the evaluator when a "past" target is proposed during planning; **2)** Trajectory-level relabeling is, by design, limited. Short trajectories, common in many training procedures, cover limited portions of the state space and prevent evalu-

ators from learning about distant targets, risking delusions when such distant targets are proposed. Short trajectories can be the product of experimental design (initial state distributions, maximum episode lengths (Erraqabi et al., 2022), or environment characteristics, *e.g.*, density of terminal states).

Avoiding feasibility delusions requires learning from all kinds of targets, including those that can never be experienced. This is to counter the exposure bias - the discrepancy between (most existing) TAP agents' behaviors (involving all targets that can be generated) and training (learning from only experienced targets), identified in Talvitie (2014).

We introduce 2 data augmentation (relabeling) strategies to expand training source-target pair distributions.

### 4.3.1. GENERATE: EXPOSE CANDIDATES TARGETS (TO BE GENERATED)

The first strategy, named GENERATE, is to *expose the targets that will be proposed during planning to the evaluator*, so that it can figure out if these targets are infeasible.

We can implement this as a Just-In-Time (JIT) relabeling strategy that relabels a sampled (un-relabeled) transition for training with a generated target (provided by the generator). We can expect GENERATE to be effective, as evaluators will get exposed to hallucinated targets that the generator could offer. Note that GENERATE requires the use of the generator, thus it incurs additional computational burden, depending on the complexity of target generation. The JIT-compatibility lowers the need for storage and provides timely coverage of the generators' changing outputs, especially helpful for non-pretrained generators. The idea behind GENERATE can be traced back to Talvitie (2014).

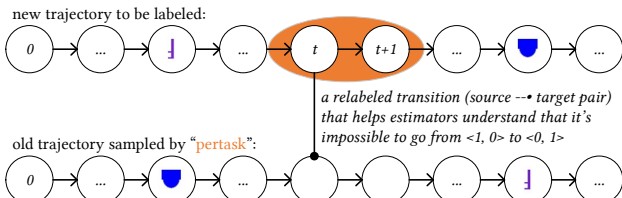

*Figure 3.* **An Example of How PERTASK Reduces E.2 by Sampling Across Episodes**: The new trajectory acquired the sword first and the shield later, while the old trajectory acquired the shield first and then the sword. When relabeling a transition in the new trajectory (in $\langle 1, 0 \rangle$), a target observation in the old trajectory (in $\langle 0, 1 \rangle$) can be paired to make an agent realize the infeasibility of the relabeled target, thus reducing E.2 delusions.

### 4.3.2. PERTASK: EXPOSE EXPERIENCED TARGETS BEYOND THE EPISODE

The second strategy, named PERTASK, is to *expose the evaluator to **all** targets $g(s^\odot)$ experienced before*, so that it could realize if some previously achieved targets not present in the current episode are infeasible from the current state.

We implement PERTASK by relabeling transitions with (the target embedding of) observations from the same training task, sampled across the entire replay. PERTASK can be seen as an generalization of the "random" strategy in Andrychowicz et al. (2017) to multi-task training settings. Importantly, PERTASK exposes the evaluator to E.2 delusions and to long-distance E.0 caused by trajectory-level relabeling on short trajectories. An example is shown in Fig. 3.

### 4.3.3. APPLICABILITY

PERTASK cannot address E.1 delusions. Meanwhile, GENERATE can be used against **some** G.2 targets that the generator hallucinates. PERTASK is a specialized and computationally efficient strategy to reduce feasibility delusions towards *all* experienced G.2 target states and importantly also the long-distance E.0 errors that GENERATE cannot handle. PERTASK is expected to be more effective than GENERATE in generalization-focused scenarios, where the distribution of G.0 & G.2 targets proposed by the generator during evaluation can go beyond those trained under GENERATE.

Importantly, relabeling strategies such as FUTURE, EPISODE and PERTASK rely on the existence of $g$ that maps a state into a target embedding, which is commonly found in TAP agents (Andrychowicz et al., 2017). However, if only the target set indicator function $h$ is available, we may need to accumulate $\langle s, \boldsymbol{g} \rangle$ tuples for which $h(s, \boldsymbol{g}) = 1$, and the use them to train a $g$. Or, in the cases where feasibility is only used for rejection, such as when dealing with simulated experiences and tree search, we could also rely on only GENERATE, which does not require $g$; Sometimes, it is $h$ that needs to be constructed. We provide detailed discussions for applying our solution on DreamerV2 in Sec. J (Appendix), with a focus on how to construct a proper $h$.

### 4.3.4. MIXTURES

Both GENERATE & PERTASK bias the training data distribution, making the evaluator spread out its learning efforts to the source-target pairs possibly distant from each other. Despite increasing training data diversity, distant pairs are less likely to contribute to better evaluation compared to the closer in-episode ones offered by EPISODE, as close-proximity G.0 targets matter the most.

Creating a mixture of sources of training data can increase the diversity of source-target combinations. For HER specifically, each atomic strategy, enumerated in Tab. 3 and illustrated in Fig. 8 (Appendix), exhibits different estimation accuracies for different types of source-target pairs, including short-distance and long-distance ones involving all of G.0, G.1 and G.2.

When the training budget is fixed, *i.e.*, training frequency, batch sizes, *etc.*, stay unchanged, the mixing proportions

of strategies pose a tradeoff to the learning of different kinds of source-target pairs. In the experiments, we mainly used (E+P+G), which is a mixture of $2/3$ EPISODE and $1/3$ PERTASK, with $1/4$ chance using GENERATE JIT, resulting in a mixture of $50\%$ EPISODE, $25\%$ PERTASK and $25\%$ GENERATE. (E+P+G) exploits the fact that assisting EPISODE with GENERATE and PERTASK often results in better performance in evaluator training, striking a balance between the investment of training budgets for the feasible and infeasible targets (Nasiriany et al., 2019; Yang et al., 2021).

### 4.4. Computational Overhead

The portion of overhead for the evaluation of targets is straightforward, as each target will be fed into the neural networks (paired with a source state) for a forward pass at inference time. This portion of the overhead depends on the complexities of evaluator's architecture and of the state / target representations. We can expect fast evaluations with lightweight evaluators.

It is the strategy post evaluation that determines the overall overhead, which depends on the planning behavior of the TAP agent that the evaluator is attached to. For background TAP agents that generate batches of targets, the improper ones can be rejected and the whole batch can be all rejected without problem (no Dyna update this time). For decision-time TAP agents, targets act as subgoals and when they are rejected, the agent can either re-generate or commit to more random explorations.

## 5. Experiments

To investigate the effectiveness and generality of our proposed solution against delusional behaviors caused by hallucinated targets, we use a simple and unified implementation of our evaluator (3-layers of ReLU activated MLP with output bin $T = 16$ and (E+P+G)) for 8 sets of experiments, encompassing decision-time *v.s.* background planning, TAP methods compatible with arbitrary $\tau$s and fixed $\tau$s, singleton and non-singleton targets, on controlled environments with respective emphases on G.1 and G.2 difficulties. The implementations for these experiments can be extended to various existing TAP methods, per Tab. 2 (in Appendix). Due to page limit, we only provide brief summaries of our experimental results in the main paper and refer the readers to the Appendix for more comprehensive details.

**Exp.**$^{1/8}$**:** Skipper (decision-time TAP with singleton targets, arbitrary $\tau$) on SSM

**Exp.**$^{2/8}$**:** (Appendix, Sec. C) LEAP (decision-time TAP with singleton targets, arbitrary $\tau$) on SSM

**Exp.**$^{3/8}$**:** (Appendix, Sec. D) Skipper on RDS, another controlled environment focusing on G.1 difficulties

**Exp.**$^{4/8}$**:** LEAP on RDS

**Exp.**[5/8]**:** Dyna (background TAP with singleton targets, $\tau = 1$) on `SSM`

**Exp.**[6/8]**:** (Appendix, Sec. F) Dyna on `RDS`

**Exp.**[7/8]**:** (Appendix, Sec. G) Feasibility estimation of *non-singleton* targets with arbitrary $\tau$ on `SSM`

**Exp.**[8/8]**:** Feasibility of *non-singleton* targets with arbitrary $\tau$ on `RDS`

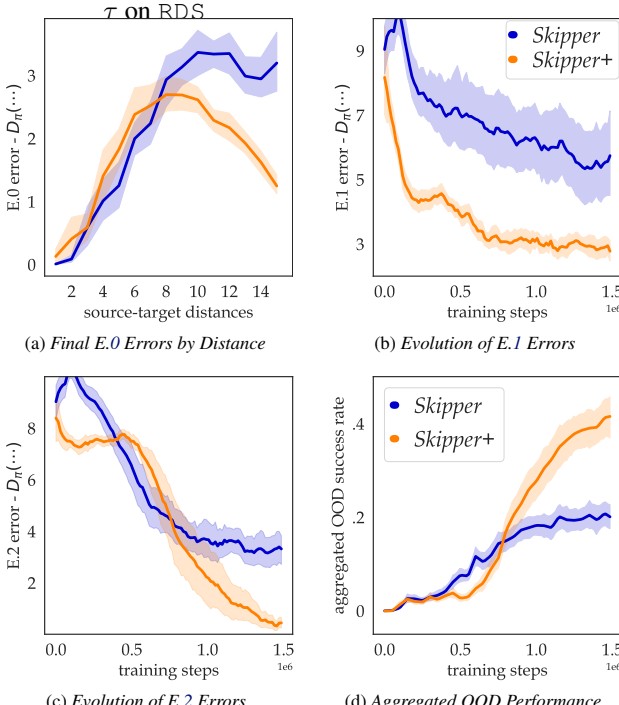

(a) *Final E.0 Errors by Distance*

(b) *Evolution of E.1 Errors*

(c) *Evolution of E.2 Errors*

(d) *Aggregated OOD Performance*

*Figure 4.* Skipper **on `SSM`**: We compare the original form of Skipper, which learns its own feasibility estimates of target states in its own way, against Skipper+, which has our proposed evaluator injected to assist the evaluation of the feasibility of targets, powered by the (E+P+G) relabeling strategy. Results of more variants are presented in Fig. 10 (Appendix). **a)**: Final E.0 errors separated across a range of ground truth distances. Both estimated and true distances are conditioned on the evolving policies; **b)**: E.1 errors measured as $L_1$ error in estimated (clipped) distance throughout training; **c)**: G.2-counterparts of **b**; **d)**: Each data point represents OOD evaluation performance aggregated over $4 \times 20$ newly generated tasks, with mean difficulty matching training. The decomposed results for each OOD difficulty are presented in Fig. 11 (Appendix).

All presented mean curves and the 95%-confidence interval bars are established over 20 independent seed runs.

## 5.1. Decision-Time Planning (Exp. [1/8] - [4/8])

For decision-time TAP agents, we are interested in understanding how rejecting hallucinated targets can influence their abilities to generalize their learned skills after learning from a limited number of training tasks. This also means, the evaluator is expected to learn to generalize its identification of infeasible targets in novel situations, by identifying the patterns of the infeasible targets.

For such experimental purpose, we use distributional shifts provided in `SSM` to simulate real-world OOD systematic generalization scenarios in evaluation tasks (Frank et al., 2009). For each seed run on `SSM`, we sample and preserve 50 training tasks of size $12 \times 12$ and difficulty $\delta = 0.4$. For each episode, one of the 50 tasks is sampled for training. Agents are trained for $1.5 \times 10^6$ interactions in total. To speed up training, we make the initial state distributions span all the non-terminal states in each training task, making trajectory-level relabeling even more problematic.

### 5.1.1. METHODS

To demonstrate the generality of our proposed solution against hallucinated targets for decision-time TAP, we apply it onto two methods utilizing targets quite differently:

Skipper (Zhao et al., 2024): generates candidate target states that, together with the current state, constitute the vertices of a directed graph for task decomposition, while the edges are pairwise estimations of cumulative rewards and discounts, under its evolving policy. A target is chosen after applying *value iteration*, *i.e.*, the values of targets are the $\mathcal{U}$ values of the planned paths.[2]

LEAP (Nasiriany et al., 2019): LEAP uses the cross-entropy method to evolve the shortest sequences of sub-goals leading to the task goal (Rubinstein, 1997). The immediate sub-goal of the elitist sequence is then used to condition a lower-level policy. Compared to Skipper, LEAP is more prone to delusional behaviors, since one hallucinated sub-goal can render a whole sub-goal sequence delusional.[3]

For Exp. [1/8] - [4/8], targets are observation-like generations, where G.1 & G.2 can be clearly identified. See the Sec. A.2 (Appendix) for more implementation details.

### 5.1.2. EVALUATIVE METRICS

**Feasibility Errors**: At each evaluation timing, we use the average errors of $\hat{\mathbb{E}}[D_\pi]$ against the ground truths as a proxy to understand the convergence of the evaluators' estimated feasibility of targets.

**Delusional Behavior Frequencies**: We monitor the frequency of a hallucinated target (made of G.1 and G.2) being chosen by the agents (as the next sub-goal for Skipper, as a part of the sub-goal chain for LEAP), *i.e.*, delusional planning behaviors. Due to the page limit, some related discussions and results are presented in Appendix.

---

[2]As shown in Tab. 2, our adaptation for Skipper can be extended to methods utilizing arbitrarily distant targets, including background TAP methods such as GSP (Lo et al., 2024)

[3]As shown in Tab. 2, our implementation for LEAP can be extended to planning methods proposing sub-goal sequences, such as PlaNet (Hafner et al., 2019)

**OOD Generalization Performance**: We analyze the changes in agents' OOD generalization performance. The evaluation tasks (targeting systematic generalization) are sampled from a gradient of OOD difficulties - 0.25, 0.35, 0.45 and 0.55. Because of the lack of space, we present the "aggregated" OOD performance, such as in Fig. 4 d), by sampling 20 task instances from each of the 4 OOD difficulties, and combine the performance across all 80 episodes, which have a mean difficulty matching the training tasks. To maximize the evaluation difficulty, the initial state is fixed in each evaluation task instance: the agents are not only spawned to be at the furthest edge to the monster, but also in semantic class $\langle 0, 0 \rangle$, *i.e.*, with neither the sword nor the shield in hand.

### 5.1.3. A GLIMPSE ON SKIPPER ON SSM (EXP. 1/8)

We compare the original form of Skipper with Skipper+, a variant that is assisted by the proposed evaluator. Details of the variants are shown in the captions of Fig. 4.

**Hallucination**: we first investigate generator's rates of hallucinations. As shown in Fig. 9 (Appendix), the generator produces targets that correspond to G.1 and G.2 with the rate of around 3% and 5%, respectively. We leave the details of the generators there for the readers.

**Feasibility Errors**: Skipper relies on a built-in cumulative discount estimator whose estimations can be converted to feasibility estimates that our evaluator seeks to learn. Thus, we can examine the errors of the feasibility estimates corrected by the injected evaluator to understand how the proposed evaluator could reduce feasibility delusions of arbitrary-horizon TAP methods. From Fig. 4 **b**) and **c**), we can see that feasibility estimates corrected by our evaluator have significantly less errors compared to the original, towards both G.1 and G.2 targets. As a perk for Skipper+'s utilization of PERTASK for E.2 delusions (included in (E+P+G)), its positive effect on far-away G.0 targets are also shown in Fig. 4 **a**). It can be seen that the evaluator is generally helpful for Skipper to understand the feasibility of all G.0, G.1 and G.2 targets.

**Frequency of Delusional Plans**: The purpose of identifying infeasible targets is to reduce delusional plans that involve them. We provide detailed results on this in Fig. 10 (Appendix), where we observed significant reduction in delusional plans involving both G.1 and G.2 targets.

**Generalization**: Comparing Skipper and Skipper+, we can deduce from Fig. 4 that generally, lower E.2 errors (**c**) lead to less frequent delusional behaviors (shown in Fig. 10, Appendix), which in turn improves the OOD performance in **d**). This indicates that rejecting infeasible targets can help decision-time TAP agents in systematic OOD generalization.

### 5.1.4. A GLIMPSE ON LEAP ON RDS (EXP. 4/8)

Having covered G.2-focused SSM with Skipper, we turn to another decision-time TAP agent compatible with arbitrary horizon targets on an environment focused on G.1 difficulties (more details in Sec. A.1, Appendix). As shown in Fig. 5, LEAP+ (LEAP assisted by the proposed evaluator), achieves significant fewer delusional plans and better OOD evaluation performance.

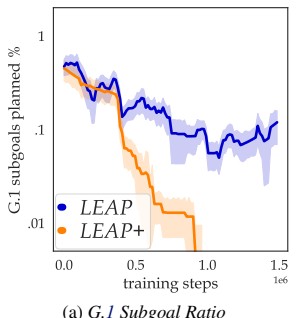
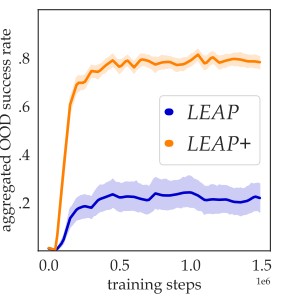

(a) *G.1 Subgoal Ratio*  (b) *Aggregated OOD Performance*

*Figure 5.* LEAP **on RDS**: compared to the baseline LEAP, LEAP+ selects significantly fewer G.1 subgoals. **a)** Evolving ratio of G.1 subgoals among the planned subgoal chains; **b)**: Aggregated OOD evaluation performance, same as for Fig.4 d).

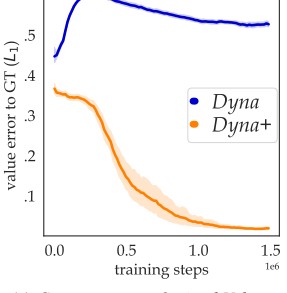
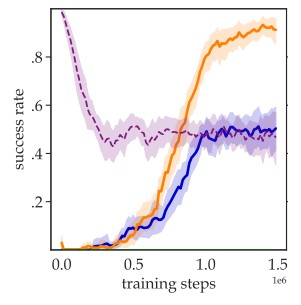

(a) *Convergence to Optimal Value*  (b) *Training Performance*

*Figure 6.* Dyna **on SSM**: compared to the baseline Dyna, Dyna+ rejects the updates toward 1-infeasible generated states flagged by the evaluator, powered by (E+P+G). **a)** Evolving mean $L_1$ distances between estimated $Q$ & optimal values; **b)**: task performance on the 50 training tasks & rate of Dyna+ rejecting updates.

### 5.1.5. SUMMARY OF EXP. 1/8 - 4/8

For Exp. 1/8 - 4/8, with the proposed evaluator, we saw a reduction in feasibility delusions and in delusional behaviors, which led to better OOD generalization performance, against challenges of G.1 & G.2. These 4 sets of experiments align in terms of the effectiveness of our approach.

### 5.2. Background Planning: A Glimpse on Exp. 5/8 & 6/8

These experiments focus on a rollout-based background TAP agent - the classical 1-step Dyna (Sutton, 1991), which uses its learned transition model to generate next states from existing states to construct simulated transitions that are used to update the value estimator, *i.e.* a "Dyna update". Jafferjee et al. (2020) demonstrated the benefit when the

delusional Dyna updates bootstrapped on hallucinated targets are rejected with an oracle. We replace the oracle using our learned evaluator. With the same training setup, in Fig. 6, we present the empirical results of how target rejection can significantly improve the performance of Dyna on SSM. The rejection rate stabilizes as both the generator and the evaluator learns. These observations are consistent with Exp. $^6/_8$, presented in Sec. F (Appendix).[4]

### 5.3. Non-Singleton Targets: A Glimpse on Exp. $^7/_8$ & $^8/_8$

We validate the evaluator's empirical convergence to ground truths when facing non-singleton targets. We present the results on RDS (Exp. $^8/_8$) in Fig. 7 and leave the SSM counterpart Fig. 19 in Appendix (for Exp. $^7/_8$). The results show convergence as expected and more details are presented in Sec. G in Appendix.

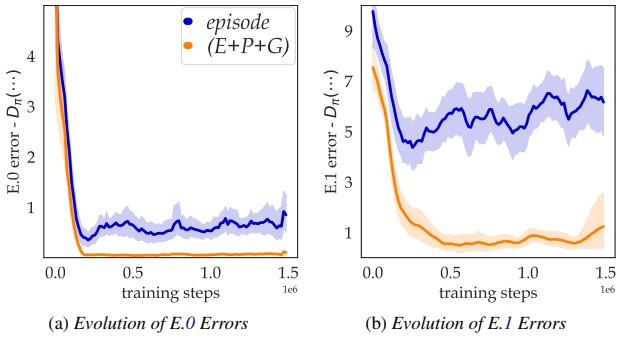

(a) *Evolution of E.0 Errors*    (b) *Evolution of E.1 Errors*

*Figure 7.* **Feasibility of Non-Singleton Targets on RDS**: **a)** Evolution of E.0 error; **b)** Evolution of E.1 error; The training data is acquired with random walk.

## 6. Related Works

**TAP**: Most rollout-based TAP methods are oblivious to model hallucinations and utilize all generated targets without question. These include fixed-step background methods such as Sutton (1991); Łukasz Kaiser et al. (2020); Yun et al. (2024); Lee et al. (2024) and decision-time methods based on tree-search, such as Schrittwieser et al. (2019); Hafner et al. (2019); Zhao et al. (2021); Zhang et al. (2024a); TAP methods compatible with arbitrarily distant targets ($\tau = \infty$) often struggle to produce non-delusional feasibility-like estimates for hallucinated targets. Thus, they cannot properly reject infeasible targets despite having their own "evaluators". These include background methods such as Lo et al. (2024) and decision-time methods for path planning (Nasiriany et al., 2019; Yu et al., 2024; Duan et al., 2024), OOD generalization (Zhao et al., 2024), and task decomposition (Zadem et al., 2024). Previously, there were method-specific approaches proposed against delusional planning behav-

iors, such as by constraining to certain probabilistic models (Deisenroth & Rasmussen, 2011; Chua et al., 2018), or training a target evaluator separately on a collected dataset.

**Delusions** in value estimates of hallucinated states are hypothesized to plague background planning (Jafferjee et al., 2020). Lo et al. (2024) introduced a temporally-abstract background TAP method to limit temporal-difference updates to only a few trustworthy targets. Langosco et al. (2022) classified goal mis-generalization, a delusional behavior describing when an agent competently pursues a problematic target. Talvitie (2017) tried to trains the model to correct itself when error is produced. Zhao et al. (2024) gave first examples of delusional behaviors caused by hallucinated targets in decision-time TAP agents.

**Hindsight Relabeling** is highlighted for its improved sample efficiency towards G.0 targets, around which most follow-up works revolved as well (Andrychowicz et al., 2017; Dai et al., 2021). However, sample efficiency is not the only concern in TAP agents, as delusions toward generated targets can cause delusional behaviors leading to other failure modes. Shams & Fevens (2022) studied the sample efficiency of atomic strategies, without looking into their failure modes. Deshpande et al. (2018) detailed experimental techniques in sparse reward settings using FUTURE. In (Yang et al., 2021), a mixture strategy similar to GENERATE improved estimation of feasible targets, though its impact on hallucinated targets was not investigated. Note that the performance of existing HER-trained agents is often limited by their reliance on FUTURE or EPISODE, whose delusions this paper intends to address.

## 7. Conclusion, Limitations & Future Work

We characterized how generator hallucinations can cause trouble for TAP agents. Then, we proposed to evaluate the feasibility of targets *s.t.* the infeasible hallucinations can be properly rejected during planning. We proposed a combination of learning rules, architectures and data augmentation strategies that leads to robust and accurate output when the proposed evaluator is applied. In experiments, we showed that the evaluator can significantly address the harm of hallucinated targets in various kinds of planning agents.

The targets we investigated in this paper are in nature, composed of "states". Meanwhile, some other planning agents propose "targets" that are in nature "actions", which do not directly correspond to reaching states, instead, for example, to maximize certain signals without providing $h$. For future work, we will investigate those agents to understand how they are impacted by hallucinations.

---

[4]The implementation here can be extended to fixed-horizon rollout agents. In the Sec. J (Appendix), we provide details on how we applied our Dyna solution to DreamerV2 (Hafner et al., 2021).

## Reproducibility Statement

The results presented in the experiments are fully-reproducible with the source code published at `https://github.com/mila-iqia/delusions`.

## Acknowledgments

Mingde "Harry" is grateful for the financial support of the FRQ (Fonds de recherche du Québec) and the collaborative efforts during his internship at RBC Borealis (formerly Borealis AI), Montréal with Tristan. We appreciate the computational resources allocated to us from Mila (Québec AI Institute) and McGill University.

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

# Appendix: Part I - Referenced Tables & Figures

*Table 2.* **Discussed Methods, Properties & How to use the Feasibility Evaluator**

| Method | TAP Category | Delusional Planning Behaviors | How Our Solution Helps | Implementation Details & Challenges |
|---|---|---|---|---|
| Dyna (Sutton, 1991) | Fixed-Horizon Background Planning | The imagined transitions could contain hallucinated (next) observations / states, whose delusional value estimates could destabilize the bootstrapping-based TD learning | Cancel updates involving rejected next states (not evaluated to be reachable within 1 timestep). | **Implemented** (for 1-step Dyna): If the output histogram of the evaluator has significant density on the bin corresponding to $t = 1$, then accept the generation, or else, reject |
| Dreamer (Hafner et al., 2025) | Fixed-Horizon Background Planning | The imagined trajectories could contain infeasible, hallucinated states | Do not let the rejected imagined states participate in the construction of update targets for the actor-critic system (See Sec.J). | **Implemented** (insufficient compute for results, Sec. J): Use the deterministic state $s$ as the state representation to feed to the evaluator (also for imagined future target states). Establish $h$ with Mahalanobis distance on the state representations and use the discount, reward and value predictions to force behavioral realism. Truncate $\lambda$-returns until infeasible imagined target states. |
| Director (Hafner et al., 2022) | Fixed-Horizon Decision-Time Planning (mainly) | The internally sampled goals may be unreachable | Reject unreachable goals and re-sample reachable ones | Similar to our implementation for Dreamer. |
| MuZero (Schrittwieser et al., 2019) | Fixed-Horizon Decision-Time Planning | The predicted states in the tree search could be unreachable hallucinations | Reject hallucinated state generations, regenerate node in tree search if necessary | Similar to our implementation for 1-step Dyna |
| SimPLe (Łukasz Kaiser et al., 2020) | Fixed-Horizon Background Planning | The predicted next observation could be an unreachable hallucination | Reject learning against the delusional estimates (potential) | Similar to our implementation for 1-step Dyna |
| Skipper (Zhao et al., 2024) | Arbitrary-Horizon Decision-Time Planning | Hallucinated subgoals could lead to decision-time planning committing to them, leading to unsafe behaviors | Use an evaluator to learn that the expected cumulative discount is 0 when aiming to reach the hallucinated subgoals. This disconnects the hallucinated subgoals from the current state in the planning | **Implemented**: diversify the source-target pairs with GENERATE and PERTASK mixtures. $\mathcal{G}$ is discrete and $h$ is a trivial comparison. |
| GSP (Lo et al., 2024) | Arbitrary-Horizon Background Planning | Hallucinated subgoals could lead to value estimation destabilization, like in Dyna. | Use output histogram of the add-on evaluator to correct the delusions by GSP's own estimators. Use the "support swap" technique. | Similar to our implementation for Skipper |
| LEAP (Nasiriany et al., 2019) | Arbitrary-Horizon Decision-Time Planning | Hallucinated subgoals could help fake a sequence of subgoals that is too good to be true and committed to during planning | Use an evaluator to learn that the expected cumulative distance is infinite when aiming to reach the hallucinated subgoals. This makes sure that subgoal sequences containing hallucinated subgoals will not be favored | **Implemented**: pay attention to the representation space of the sub-goals. |
| PlaNet (Hafner et al., 2019) | Arbitrary-Horizon Decision-Time Planning | Hallucinated subgoals could help fake a sequence of subgoals that is too good to be true and committed to during planning | Reject the delusional subgoals and therefore reject the delusional subgoal sequences | Same as our implementation for LEAP (both uses CEM for planning (Rubinstein, 1997)) |

*Similar colors are used to denote similar implementations for the solution proposed in this work.*

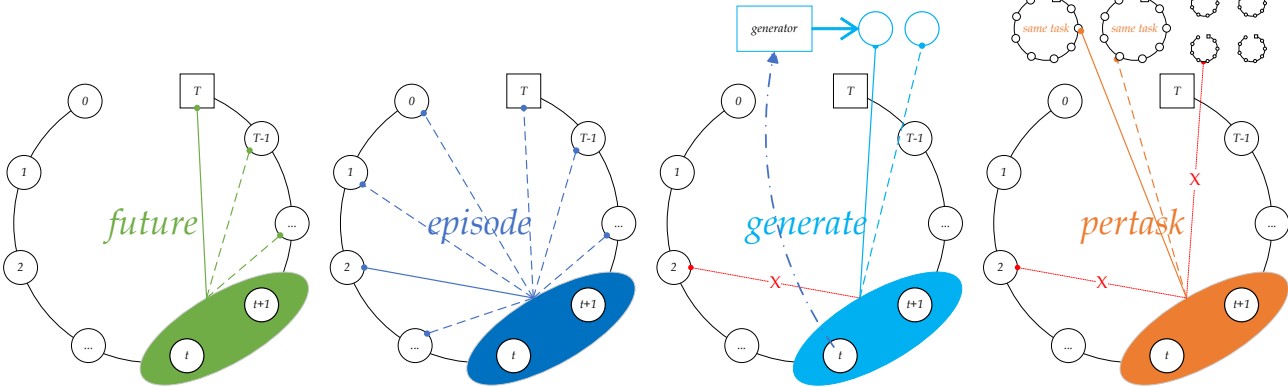

*Figure 8.* **Representative Atom Hindsight Relabeling Strategies & Newly Proposed Ones**: *The first two strategies,* FUTURE *and* EPISODE, *are widely used as they create relabeled transitions that help evaluators efficiently handle G.0 targets during planning. The last two,* GENERATE *and* PERTASK, *are effective at addressing delusions, making them useful in specific scenarios. Atomic hindsight strategies from the first group can serve as backbones for mixture strategies, complemented by the second group to address delusions.*

| Strategies | Advantages | Disadvantages | Gist |
|---|---|---|---|
| EPISODE | Efficient for evaluator to learn close-proximity relationships | When used exclusively to train evaluator, 1) cannot handle E.2 and 2) prone to E.0 - cannot learn well from short trajectories; Can cause G.2 targets when used to train generators | Creates training data with source-target pairs sampled from the same episodes |
| FUTURE | Can be used to learn a conditional generator with temporal abstractions | In addition to the shortcomings of EPISODE (those for evaluators only), this additionally causes E.0 when used as the exclusive strategy for evaluator training | Creates training data with temporally ordered source-target pairs from the same episodes |
| GENERATE | Addresses E.1 with data diversity (also E.2 when generator produces G.2) | Relies on the generator with additional computational costs; Potentially low efficiency in reducing E.0. | Augments training data to include candidate targets proposed at decision time |
| PERTASK | Addresses evaluator delusions (E.2 & E.0 for long-distance pairs) | low efficiency in learning close-proximity source-target relationships | Augments training data to include targets that were experienced |

*Table 3.* **Hindsight Relabeling Strategies**: EPISODE *and* FUTURE *are widely used as they increase sample efficiency towards G.0 states significantly;* GENERATE *and* PERTASK, *proposed in this paper, should be applied against delusions in relevant scenarios.*

# Appendix: Part II - Experiments

## A. More Details on Decision-Time TAP Experiments (Exp. $^1/_8$ - $^4/_8$)

### A.1. `RandDistShift`

The second environment employed is `RandDistShift`, abbreviated as `RDS`. `RDS` was originally proposed in Zhao et al. (2021) as a variant of the counterparts in the MiniGrid Baby-AI platform (Chevalier-Boisvert et al., 2023), and then later used as the experimental backbone in Zhao et al. (2024). `SSM` was inspired by `RDS`. We can view `RDS` as a sub-task of `SSM`, where everything happens in semantic class $\langle 1, 1 \rangle$, *i.e.*, agents always spawn with the sword and the shield in hand, thus can acquire the terminal sparse reward by simply navigating to the goal. `RDS` instances thus have smaller state spaces than its `SSM` counterparts. The most important difference, in the views of this work, is that `RDS` removed the challenges introduced by temporary infeasibility. This means that G.2 and E.2 are no longer relevant, shifting the dominance towards G.1 + E.1 combination. Using `RDS` not only showcase the performance of the proposed strategies on a controlled environment with G.1 + E.1 dominance, contrasting the G.2 + E.2 dominance of `SSM`, it also can be used to validate the performance of our adapted agents, on an environment where previous benchmarks exist.

### A.2. Generator Hallucinations

We use hindsight-relabeled transitions to train the generators in the two methods (Skipper & LEAP), to demonstrate how different ways of training the generator could affect the rates of hallucinations. G.2 can appear more frequently if the generator is trained to imagine more diverse kinds of targets than needed. For example, a conditional target generator which learns from ᴇᴘɪsoᴅᴇ will be more likely to produce G.2 targets (compared to ꜰuᴛuʀᴇ). This was why we mostly used ꜰuᴛuʀᴇ to train the generators in the related experiments.

For the HER-trained generators, Fig. 9 **a)**, shows that different training targets for the generator could lead to different degrees of hallucinations, in terms of G.1 and G.2. Importantly, Fig. 9 **b)** indicates that, ꜰuᴛuʀᴇ generates G.2 targets significantly less frequently than ᴇᴘɪsoᴅᴇ and ᴘᴇʀᴛᴀsᴋ, as the other two wasted training budget on G.2 targets, especially ᴘᴇʀᴛᴀsᴋ that brings in more problematic training samples from long distances. *In all other experiments, we only compare variants with* ꜰuᴛuʀᴇ *for the generator training.*

The generator is consistently used for both Skipper and LEAP in Exp. $^1/_8$ - Exp. $^4/_8$.

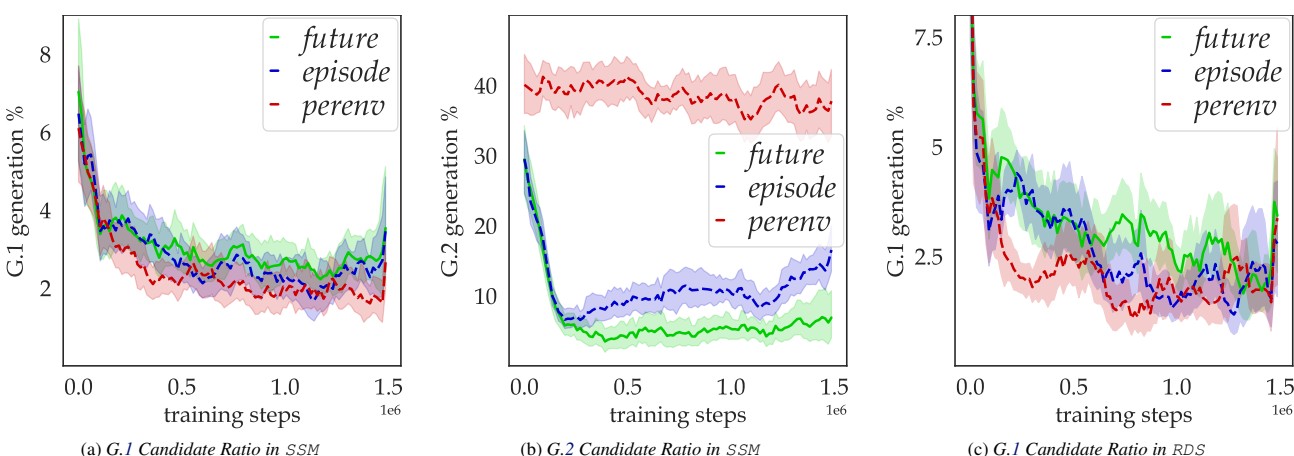

(a) *G.1 Candidate Ratio in* `SSM`          (b) *G.2 Candidate Ratio in* `SSM`          (c) *G.1 Candidate Ratio in* `RDS`

*Figure 9.* **Hallucination Frequencies**: **a)** Evolving ratio of G.1 "states" among all candidates at each target selection, throughout training; Subfigure **b)** is the E.2-counterpart of **a)** on `SSM`; Subfigure **c)** is the `RDS`-counterpart of **a)**.

## B. Skipper on **SSM** (Exp. $^1/_8$, continued)

### B.1. Additional Results of Skipper+ with other Relabeling Strategies & Frequencies of Delusional Plans

In the main manuscript, we focused on a particular implementation of evaluator which utilizes (E+P+G) for training data. Since we have the full degrees of freedom in deciding the mixture ratios of the involved relabeling strategies, *i.e.*, EPISODE, GENERATE & PERTASK, we will provide more results here that encompass more relabeling strategies. These results could not only provide the readers with more understanding of the empirical characteristics of the relabeling strategies but also can serve as an ablation test for the two alternative relabeling strategies, *i.e.*, GENERATE & PERTASK. The variant relabeling strategies are as follows:

- **(E+G)** - a mixture against E.1. EPISODE with $50\%$ chance using GENERATE JIT, resulting in a half-half mixture of EPISODE & GENERATE

- **(E+P)** - against E.2. Half EPISODE & half PERTASK

We can see that (E+P+G) is the middle ground between (E+G) and (E+P), with a comprehensive coverage for both G.0, G.1 and G.2 cases. This is why we have chosen (E+P+G) as the default for our evaluator, since we do not wish to assume access to the state space structures of the environments. Note that for your convenience, we have used consistent colors for each variant throughout this work.

We expand the results in Fig. 4 to Fig. 10, to not only include new sub-figures on the frequencies of delusional planning behaviors but also the variants.

From Fig. 10 a) and d), we can see that the more relabeling is invested into GENERATE, the less E.1 errors the agents would have and the less frequent G.1 targets trigger delusional plans. The same can be said for G.2 & PERTASK in Fig. 10 b) and e). Because of the state space structure, on SSM, it is quite expected that (E+P) is the most efficient in terms of increasing OOD evaluation performance (Fig. 10 f)) due to the dominating challenge of G.2 targets.

### B.2. Breakdown of Task Performance

In Fig. 11, we present the evolution of Skipper variants' performance on the training tasks as well as the OOD evaluation tasks throughout the training process. Note that Fig. 10 **f)** (and by extension Fig. 4 **d)**) is an aggregation of all 4 sources of OOD performance in Fig. 11 **b-e)**.

From the performance advantages of the hybrid variants (in both training and evaluation tasks), we can see that learning to address delusions during training brings better understanding for novel situations posed in OOD tasks.

## C. LEAP on **SSM** (Exp. $^2/_8$)

This set of experiments seeks to demonstrate that the proposed feasibility evaluator can help reduce delusional planning behaviors in other decision-time TAP agents while facing challenges of G.2 targets. For this purpose, we study LEAP performance on SSM, and its variant LEAP+ with our evaluator injected.

Compared to Skipper, LEAP utilizes the generated targets in a very different way, as its decision-time planning process constructs a singular sequence of subgoals leading to the task goal. Due to a lack of backup subgoals, even if one among them is problematic, the whole resulting plan would be delusional, making LEAP much more prone to failures compared to Skipper, where candidate targets can still be reused if deviation from the original plan occurred.

SSM has a relatively large state space that requires more intermediate subgoals for LEAP's plans. However, an increment of the number of subgoals also dramatically increases the frequencies of delusional plans, damaging the agents' performance. Because of this, our experimental results of LEAP on SSM with size $12 \times 12$ became difficult to analyze because of the rampant failures. We chose instead to present the results on SSM with size $8 \times 8$ here.

Additionally, consistent with Exp. $^1/_8$, for a more comprehensive understanding of the relabeling strategies' impact on the learned evaluator, we include results beyond (E+P+G), which is the default used in the main paper.

For LEAP, we use some different metrics to analyze the effectiveness of the proposed strategies in addressing delusions. This is because, if LEAP's evaluator successfully addressed delusions and learned not to favor the problematic targets (G.1

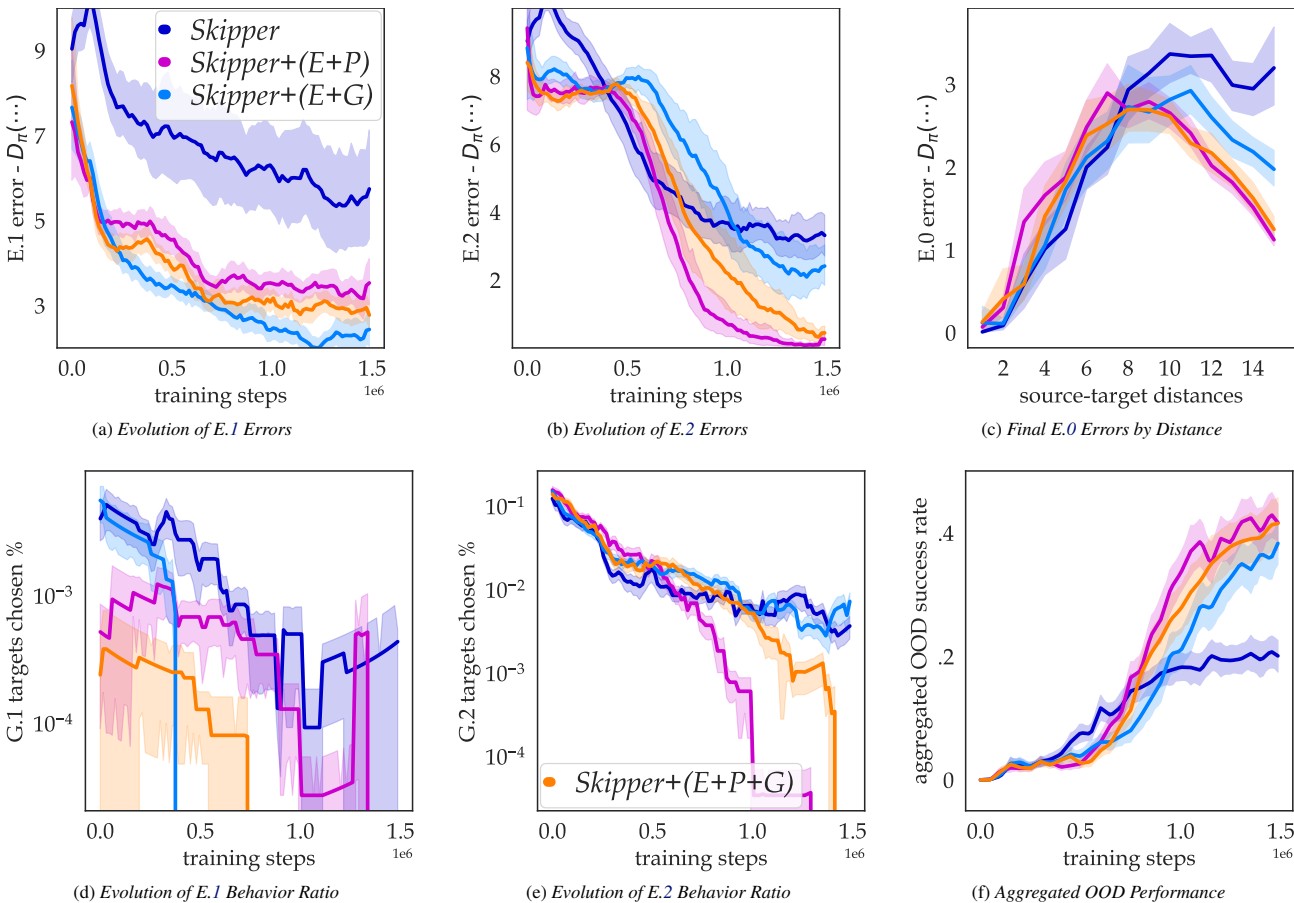

(a) *Evolution of E.1 Errors*     (b) *Evolution of E.2 Errors*     (c) *Final E.0 Errors by Distance*

(d) *Evolution of E.1 Behavior Ratio*     (e) *Evolution of E.2 Behavior Ratio*     (f) *Aggregated OOD Performance*

*Figure 10.* Skipper **and More Variants of** Skipper+ **on** `SSM`: In addition to subfigures that already exist in Fig. 4, *i.e.*, **a)**, **b)**, **c)**, & **f)**, we provide additional subfigures **d)** and **e)**, to demonstrate the changes of frequencies in delusional behaviors throughout training, for G.1 and G.2 composed targets, respectively. The curves denote the frequencies of G.1 and G.2 targets becoming the imminent subgoals that Skipper seeks to achieve next. Each figure is augmented with results of additional evaluator variants with different relabeling strategies.

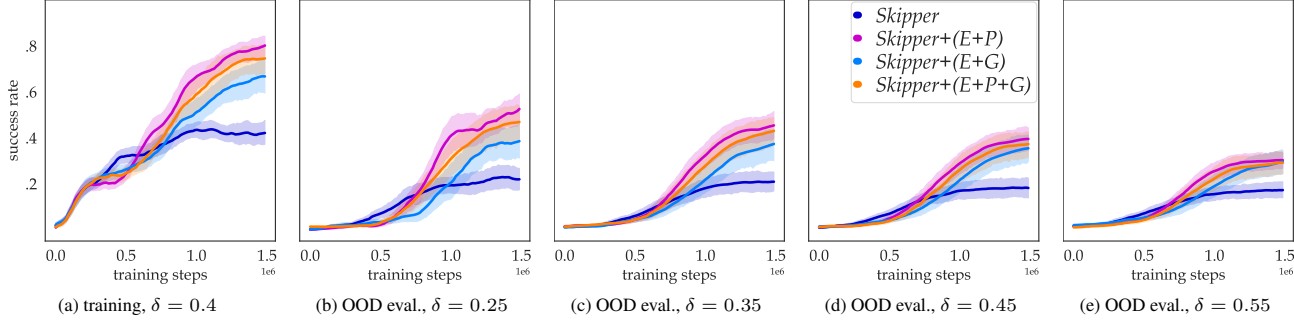

(a) training, $\delta = 0.4$    (b) OOD eval., $\delta = 0.25$    (c) OOD eval., $\delta = 0.35$    (d) OOD eval., $\delta = 0.45$    (e) OOD eval., $\delta = 0.55$

*Figure 11.* **Separated Evolution of OOD Performance of** Skipper **Variants on** `SSM`

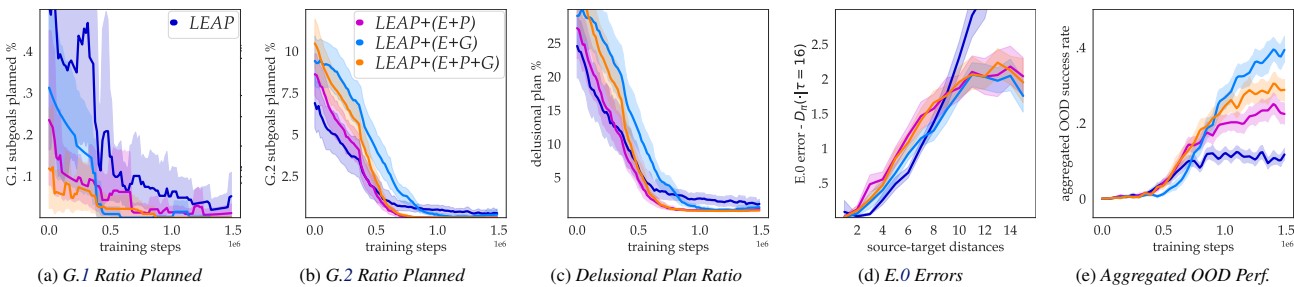

(a) *G.1 Ratio Planned*    (b) *G.2 Ratio Planned*    (c) *Delusional Plan Ratio*    (d) *E.0 Errors*    (e) *Aggregated OOD Perf.*

*Figure 12.* LEAP **on SSM**: **a)** Ratio of G.1 subgoals among the planned sequences; **b)** Ratio of G.2 subgoals in the planned sequences; **c)** Ratio of evolved sequences containing at least one G.1 or G.2 target; **d)** The final estimation accuracies towards G.0 targets after training completed, across a spectrum of ground truth distances. In this figure, both distances (estimation and ground truth) are conditioned on the final version of the evolving policies; **e)** Each data point represents OOD evaluation performance aggregated over $4 \times 20$ newly generated tasks, with mean difficulty matching the training tasks.

and G.2), then they will not be selected in the evolved elitist sequence of subgoals. This makes it inconvenient for us to use the distance error in the delusional source-target pairs during decision-time as a metric to analyze the reduction of delusional estimates, because of their growing scarcity.

As we can see from Fig. 12, similar arguments about the effectiveness of the proposed hybrid strategies can be made, to those with Skipper. The hybrids with more investment in addressing E.1, *i.e.*, (E+G) and (E+P+G), exhibit the lowest E.1 errors (**a**)). Similarly, (E+P) and (E+P+G) achieve the lowest E.2 errors (**b**)). In **e**), we see that the 3 hybrid variants achieve better OOD performance than the baseline E. Specifically, (E+G) achieved the best performance. This is likely because that it induced the highest sample efficiency in terms of learning the estimations towards G.0 subgoals, as shown in **d**). Assistive strategies such as GENERATE and PERTASK do not only induce problematic targets, but also G.0 ones that can shift the training distribution towards higher sample efficiencies in the traditional sense.

### C.0.1. BREAKDOWN OF TASK PERFORMANCE

In Fig. 13, we present the evolution of LEAP variants' performance on the training tasks as well as the OOD evaluation tasks throughout the training process. Note that Fig. 12 **e)** is an aggregation of all 4 sources of OOD performance in Fig. 13 **b-e)**.

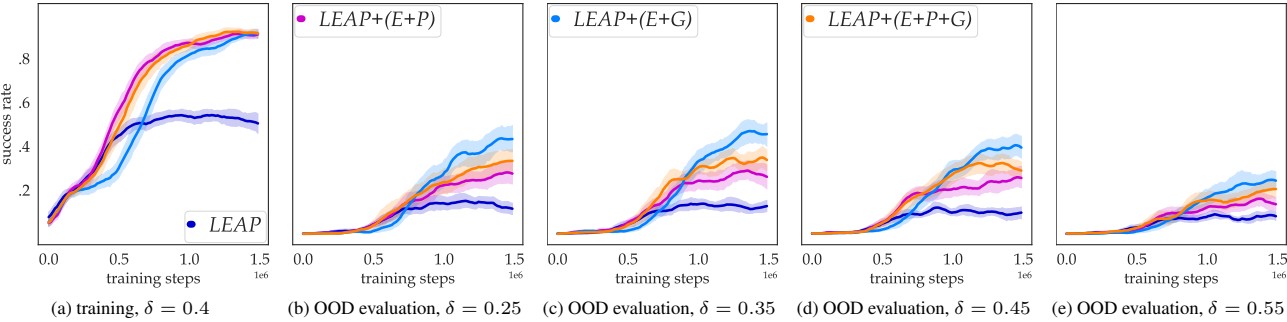

(a) training, $\delta = 0.4$    (b) OOD evaluation, $\delta = 0.25$    (c) OOD evaluation, $\delta = 0.35$    (d) OOD evaluation, $\delta = 0.45$    (e) OOD evaluation, $\delta = 0.55$

*Figure 13.* **Evolution of OOD Performance of** LEAP **Variants on SSM**

## D. Skipper **on RDS (Exp.** $3/8$**)**

This set of experiments focus on the feasibility evaluator's abilities in the face of G.1 challenges. We present Skipper's evaluative curves in Fig. 14.

From Fig. 14 **d)**, we can see that, probably because of the lack of dominant G.2 + E.2 cases, the OOD performance of even the baseline Skipper is high (compared to the performance of LEAP baseline, because Skipper's planning behaviors are less prone to infeasible targets). (E+G), *i.e.* the hybrid with the most investment in GENERATE (aiming at E.1), performs the

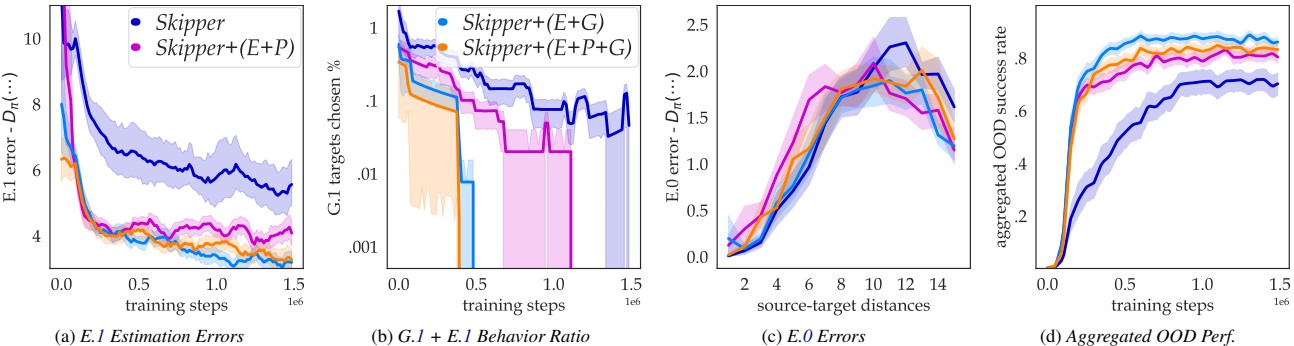

*Figure 14.* Skipper **on RDS**: **a)** E.1 delusions in terms of $L_1$ error in estimated distance is visualized, throughout the training process. **b)** The curves represent the frequencies of choosing G.1 "states" whenever a selection of targets is initiated; **c)** The final estimation accuracies towards G.0 targets after training completed, across a spectrum of ground truth distances. In this figure, both distances (estimation and ground truth) are conditioned on the final version of the evolving policies; The state structure of RDS does not permit G.2 targets and the corresponding E.2 delusions; **d)** Each data point represents OOD evaluation performance aggregated over $4 \times 20$ newly generated tasks, with mean difficulty matching the training tasks.

best both in terms of E.1 delusion suppression (**a**)), and OOD generalization (**d**)), as expected. Also, we observe similarly that the more the evaluator is invested in GENERATE, which is appropriate for RDS without G.2 challenges, the higher the performance.

### D.1. Breakdown of Task Performance

In Fig. 15, we present the evolution of Skipper variants' performance on the training tasks as well as the OOD evaluation tasks throughout the training process. Note that Fig. 14 **d)** is an aggregation of all 4 sources of OOD performance in Fig. 15 **b-e)**.

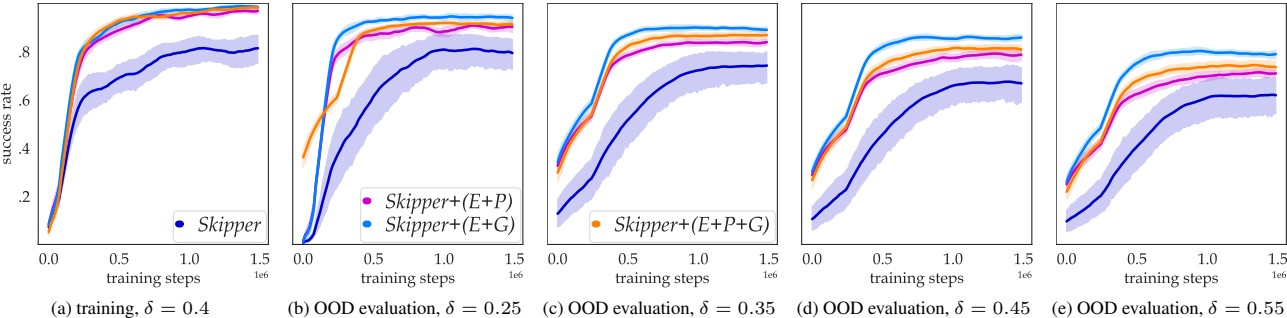

*Figure 15.* **Evolution of OOD Performance of** Skipper **Variants on RDS**

## E. LEAP on RDS (Exp. $4/8$)

The last set of experiments focus on LEAP's performance on RDS. Previously, in the main manuscript, we briefly looked at the results of this part without variants other than (E+P+G).

Similarly, we present the evaluative metrics in Fig. 16.

The results in this set of experiments (Exp. $4/8$) are very similar to that of Skipper on RDS (Exp. $3/8$).

The conclusions are similar, despite that the OOD performance gain by addressing delusions is significantly higher than in SSM.

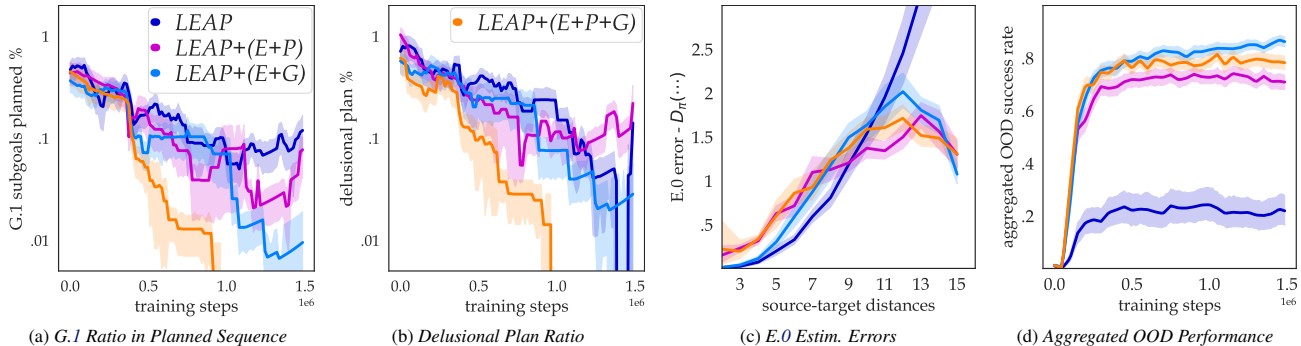

(a) *G.1 Ratio in Planned Sequence*    (b) *Delusional Plan Ratio*    (c) *E.0 Estim. Errors*    (d) *Aggregated OOD Performance*

*Figure 16.* LEAP **on RDS**: **a)** Ratio of G.1 subgoals among the planned sequences; **b)** Ratio of planned sequences containing at least one G.1 target; **c)** The final estimation accuracies towards G.0 targets after training completed, across a range of ground truth distances. In this figure, both distances (estimation and ground truth) are conditioned on the final version of the learned policies; **d)** Each data point represents OOD evaluation performance aggregated over $4 \times 20$ newly generated tasks, with mean difficulty matching the training tasks.

### E.0.1. BREAKDOWN OF TASK PERFORMANCE

In Fig. 17, we present the evolution of LEAP variants' performance on the training tasks as well as the OOD evaluation tasks throughout the training process. Note that Fig. 16 **d)** is an aggregation of all 4 sources of OOD performance in Fig. 17 **b-e)**.

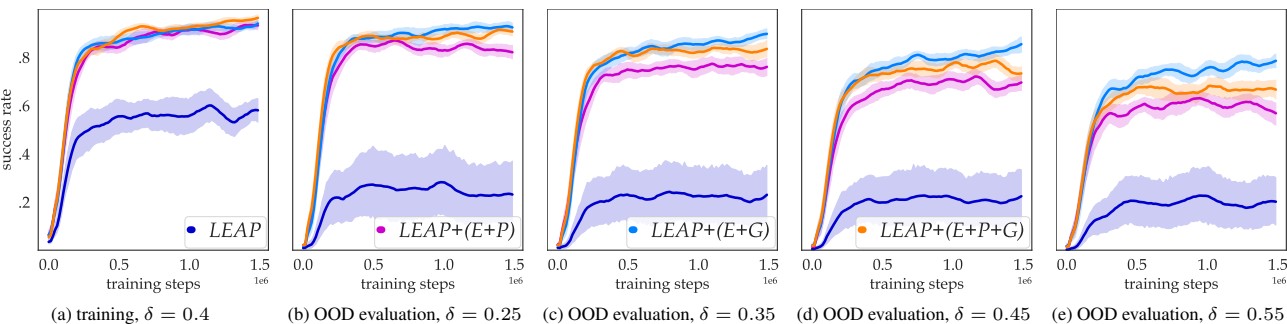

(a) training, $\delta = 0.4$    (b) OOD evaluation, $\delta = 0.25$    (c) OOD evaluation, $\delta = 0.35$    (d) OOD evaluation, $\delta = 0.45$    (e) OOD evaluation, $\delta = 0.55$

*Figure 17.* **Evolution of OOD Performance of** LEAP **Variants on RDS**

## F. Background Planning: Dyna **on RDS (Exp.**$6/8$**)**

In Fig. 18, we present the empirical performance of a Dyna variant with rejection enabled by (E+P+G), which is significantly better than the baseline.

## G. Feasibility of Non-Singleton Targets (Exp. $7/8$ & $8/8$)

For this set of experiments, we want to demonstrate the capability of the learned feasibility evaluator facing non-singleton targets.

We test if our implemented feasibility evaluator for Exp. $1/8$ - Exp. $4/8$ could withstand targets that are non-singleton. In its previous implementation, we use $h$ to enforce the that the targets are singletons. In fact, each $g^{\odot}$ takes the form of a state representation and $h$ is only activated if a state with exactly the same representation is reached. For the non-singleton experiments however, we let $h$ activate when a state is within distance one to the target state, effectively expanding each target set from size 1 to maximally size 5. Given the new termination mechanisms enforced by the new $h$, each target now, despite still taking the form of a state representation, has a new meaning. This setting mirrors the goal-conditioned path planning agents that seeks to reach certain neighborhoods of the planned waypoints.

With this setting, we can also intuitively analyze the composition of the target set. Specifically, if one of the member

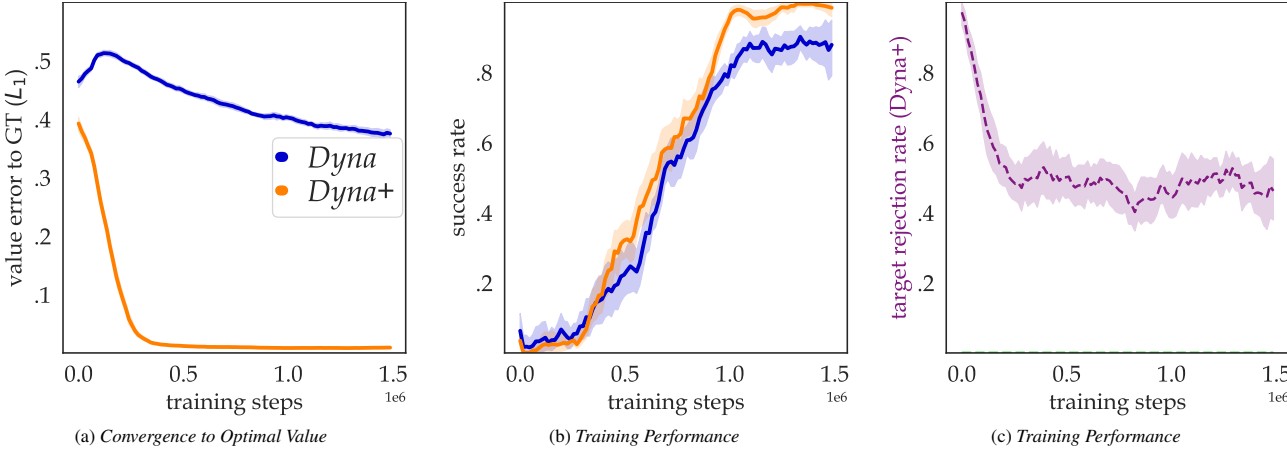

(a) *Convergence to Optimal Value*    (b) *Training Performance*    (c) *Training Performance*

*Figure 18.* Dyna **on RDS**: **a)**: Evolving mean $L_1$ distances between estimated $Q$ values & ground truth optimals; **b)**: evaluation performance on the 50 training tasks; **c)**: rate of rejecting Dyna updates.

state is G.2, then the whole target set are fully made of G.2. If all the 5 states are out of the state space, then the target is fully composed of G.1. For SSM, a target in the temporarily unreachable situation, *e.g.*, $s \in \langle 1, 1 \rangle$ with target encoding $s^\odot \in \langle 0, 1 \rangle$, could be composed of not only G.2 states but also some G.1.

We apply the new $h$ to evaluator training and to the ground truth DP solver, and then compare their differences. As we could observe from Fig. 19, the proposed feasibility evaluator, with the help of the two assistive hindsight relabeling strategy, significantly reduces the feasibility errors in all categories.

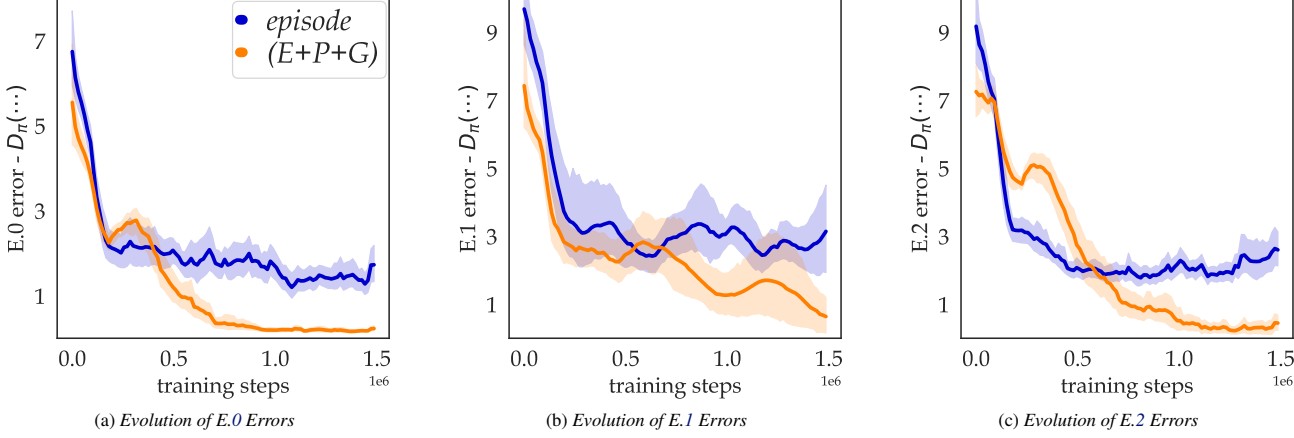

(a) *Evolution of E.0 Errors*    (b) *Evolution of E.1 Errors*    (c) *Evolution of E.2 Errors*

*Figure 19.* **Feasibility of Non-Singleton Targets on SSM**: **a)** Evolution of E.0 error; **b)** Evolution of E.1 error; **c)** Evolution of E.2 error; The training data is acquired with random walk, since the introduced non-singleton targets do not lead to adequate performances.

We observe the similar results in RDS, which was presented in Fig. 7 in the main manuscript.

As extensively discussed, our evaluator is featured with three components: 1) the off-policy compatible learning rule, 2) the unified architecture with distributional output head that can learn up to $T$-feasibility and 3) two proper relabeling strategies against feasibility delusions. In the Exp. $1/8$ - Exp. $4/8$, we demonstrated that such combination is effective against infeasible targets and we used the difference compared to the ground truth feasibility values to quantitatively measure the convergence (the reduction in feasibility errors and delusions). In these last two sets of experiments, we tried to do the same while removing the need of control, focusing on the convergence itself under a random exploration policy, to demonstrate our evaluator's general applicability.

# Appendix: Part III - Technical Details & Discussions

## H. Discussions & More Details of GENERATE & PERTASK

### H.1. Implementation of PERTASK

PERTASK takes the advantage of the fact that training is done on limited number of fixed task instances. We give each task a unique task identifier. At relabeling time, PERTASK samples observations among all the transitions marked with the same identifier as the current training task instance. This can be trivially implemented with individual auxiliary experience replays that store only the experienced states with memory-efficient pointers to the buffered $x_t$'s in the main HER.

### H.2. Discussions

GENERATE not only creates targets with G.1 "states", but also generate valid targets that should resemble the distribution it was trained on. Thus, it is not clear if mixing in data augmented by GENERATE would result in lower sample efficiency in the estimation cases involving valid targets. Take SSM as an example, GENERATE seemed to have detrimental effect to E.0 cases when applied to Skipper, while it greatly boosted accuracies for LEAP overall.

In some experiments, PERTASK demonstrated clear effectiveness in addressing E.1 as well, despite that it was not designed to. This is likely because of some generalization effects of the evaluator, which were trained with additional data that boosted data diversity.

In some environments, we expect that PERTASK could also be used (for mixtures of the generator) to learn to generate longer-distance targets from the current states if the generator has trouble doing so with FUTURE, with the accompanied risks of lower efficiency and G.2 hallucinations.

## I. Implementation Details for Experiments

### I.1. Skipper

Our adaptation of Skipper over the original implementation[5] in Zhao et al. (2024) is minimal. We have additionally added two simple vertex pruning procedures before the vertex pruning based on $k$-medoids. These two procedures include: 1) prune vertices that are duplicated, and 2) prune vertices that cannot be reached from the current state with the estimated connectivity.

We implemented a version of generator that can reliably handle both RDS and SSM with the same architecture. Please consult `models.py` in the submitted source code for its detailed architecture.

For SSM instances, since the state spaces are 4-times bigger than those of RDS, we ask that Skipper generate twice the number of candidates (both before and after pruning) for the proxy problems.

All other architectures and hyperparameters are identical to the original implementation.

For better adaptability during evaluation and faster training, Skipper variants in this paper keeps the constructed proxy problem for the whole episode during training and replanning only triggers a re-selection, while during evaluation, the proxy problems are always erased and re-constructed.

The quality of our adaptation of the original implementation can be assured by the fact the E variant's performance matches the original on RDS.

### I.2. LEAP

LEAP's training involves two pretraining stages, that are, generator pretraining and evaluator (a distance estimator) training.

We improved upon the adopted discrete-action space compatible implementation of LEAP (Nasiriany et al., 2019) from Zhao et al. (2024). We gave LEAP additional flexibility to use fewer subgoals along the way to the task goal if necessary. Also, we improved upon the Cross-Entropy Method (CEM) (Rubinstein, 1997), such that elite sequences would be kept

---

[5]https://github.com/mila-iqia/Skipper

intact in the next population during the optimization process. We increased the base population size of each generation to $512$ and lengthened the number of iterations to $10$.

For RDS $12 \times 12$ and SSM $8 \times 8$, at most 3 subgoals are used in each planned path. We find that employing more subgoals greatly increases the burden of CEM and lower the quality of the evolved subgoal sequences, leading to bad performance that cannot be effectively analyzed.

We used the same generator architecture and hyperparameters as in Skipper. All other architectures and hyperparameters remain unchanged.

Similarly for LEAP, for better adaptability during evaluation, the planned sequences of subgoals are always reconstructed whenever planning is triggered. While in training, the sequence is reused and only a subgoal selection is conducted.

The quality of our adaptation of the original implementation can be assured by the fact the E variant's performance matches the original on RDS.

### I.3. Dyna

The generator is a one-step model built for MiniGrid observations. For each batch update based on real, experienced transitions, an equal sized batch of simulated transitions will be generated with the help of the generator.

The threshold for 1-feasibility based rejections are set to be $0.05$, *i.e.*, if the feasibility estimator estimates that there is less than $5\%$ probability that a generated target state is 1-feasible, the associated update would be rejected by setting its corresponding error to be $0$ within the generated minibatch.

## J. Applying the Evaluator on Dreamerv2

To demonstrate that our approach functions effectively in more generalist settings, such as those with continuous state and action spaces and partial observability, and to illustrate its application to a modern TAP agent, we integrated our proposed evaluator into Dreamerv2 (Hafner et al., 2021). The evaluator filters out potentially delusional values from infeasible states that might distort the $\lambda$-returns derived from imagined trajectories. Given the technical complexity ahead, we suggest readers familiarize themselves with Dreamerv2 before continuing (Hafner et al., 2021).

Although Dreamerv2's stochastic states are discrete and could theoretically support similarity assessments, their design ensures they rarely repeat due to numerous possibilities, making them too random for our similarity function $h$. Consequently, we rely on the deterministic state representations $s$, which also prompt us to more thought-provoking discussions.

Dreamerv2 operates as a Dyna-like method, employing fixed-horizon rollouts with autoregressively imagined states as targets. Lacking a built-in similarity function $h$, it provides an opportunity to showcase how we construct $h$ in our approach. Our method incorporates various realism aspects to assess state similarity between the next state and the target state, influencing the branching in Eq. 2 during evaluator updates (Russell et al., 2025).

### J.1. How to craft $h$: Observational Realism

Observational realism, *i.e.*, the similarity in terms of state representations is the first obvious criteria for $h$.

Theoretically, one might simplistically assume state equivalence by defining an $\epsilon$-ball around the target state. However, in practice, an $\epsilon$-ball based on $L_2$ distances proves inadequate due to varying representation scales. Instead, we employ Mahalanobis distances, which better accommodate the representations' distributional variations.

To be more precise, we use an Exponential Moving Average (EMA) of the covariance of concatenated current-next deterministic state pairs $[s_t, s_{t+1}]$ to calculate the Mahalanobis distances between the next state pair $[s_t, s_{t+1}]$ and target state pair $[s_t, \hat{s}_{t+1}]$.

### J.2. How to craft $h$: Behavioral Realism

The second focus is behavioral realism: does the agent exhibit similar behavior (*e.g.*, in value, reward, and discount estimations) across the states ($s_{t+1}$ & $\hat{s}_{t+1}$)?

Here, we apply Mahalanobis distances to pairs of current and future values, rewards, and discounts, ensuring the states

appear similar from the agent's perspective.

Caution is required with action-realism. Naively applying our method to one-hot encoded discrete actions could result in a singular covariance matrix for the $\epsilon$-ball computation.

Preliminary Atari experiments suggest setting distinct $\epsilon$ values for different components—state representations, value estimations, reward estimations, and discount estimations.

### J.3. How to Relabel: Just-In-Time (JIT) Construction

Since Dreamerv2 samples sub-trajectories and computes state representations autoregressively, we forgo a separate HER for storing source-target pairs, opting instead for Just-In-Time (JIT) construction. Designed for single-environment training and evaluation, Dreamerv2 allows us to implement a (E+G) variant on agent-sampled sub-trajectories. Initial tests indicate a balanced mix of EPISODE (within sub-trajectories) and GENERATE performs effectively.

### J.4. How to Reject: Three Criteria for $\lambda$-returns

Dreamerv2 leverages its model to imagine future states and values, using these, along with intermediate rewards and discounts, to compute $\lambda$-returns for each origin state.

For such strategy, we implemented the following 3 criteria for rejecting the imagined states:

1. **Transition-wise Rejection**: If a next state seems unlikely to follow from the current state, its value is deemed untrustworthy. This process is repeated for all imagined transitions. Notably, a state rejected as infeasible in one transition might still be reachable elsewhere, so subsequent states are not automatically discarded.

2. **Point-to-Point (P2P) Rejection for Targets**: Starting from a replay-sampled base state, we assess whether each imagined state is reachable, regardless of steps taken. This counters hallucinated targets from accumulated errors over the imagination horizon (Talvitie, 2017), excluding such states from value estimation targets.

3. **P2P Rejection for Current States**: Entirely unreachable states are excluded as current states in multi-step value updates, though subsequent states may remain viable.

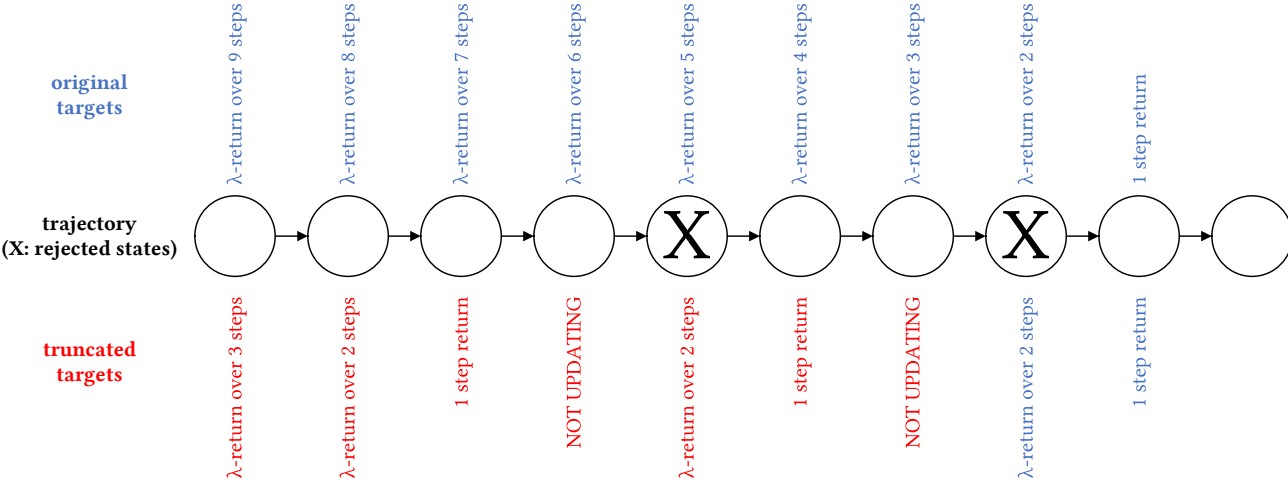

*Figure 20.* **Truncated $\lambda$-Returns with Rejected States**: the original $\lambda$-returns are illustrated in the top row, while the truncated returns are illustrated in the bottom row with the differences marked in red. Our strategy ensures that the critic targets in the trajectories can be maximally preserved for updates. The states right before the rejected ones will have no trustworthy critic targets and are thus not updated. Starting from the last rejected state, all critic targets remain the same as the originals.

The first two criteria yield a binary mask to truncate $\lambda$-returns in sampled sub-trajectories, excluding untrustworthy values

while preserving horizons for reliable ones. Our repository offers an efficient implementation, maintaining the complexity as the original, un-truncated $\lambda$-returns. The behavior of the (critic) target-based rejection is presented in Fig. 20.

The third criterion masks updates to wholly infeasible imagined states. By examining the rejection rate by the horizon index, the evaluator can also be used to understand how long the imagined trajectories are likely to be trustworthy and thus adjust the associated hyperparameters.

We developed a standalone, user-friendly evaluator (implemented in PyTorch) that integrates seamlessly into TAP agents like Dreamerv2, employing its own optimizer and target networks for robust learning when activated. Please check `evaluator.py` in the source code (`https://github.com/mila-iqia/delusions`).

We tuned the hyperparameters using the Atari environments and found that both the autoregressive estimations of distances and the P2P distances (towards the target states) in the sampled and imagined trajectories roughly converge to the estimated ground truth values, which are deduced from their time indices. This is the best we can do for environments without ground truth access.

Regrettably, our Atari100k preliminary results with $10^5$ interactions show negligible performance gains over the baseline (Łukasz Kaiser et al., 2020). This is likely because the state representations of Dreamer usually takes a significant portion of training to stabilize and for the evaluator to adapt to. The differences are expected to show with prolonged experiments where a significant number of updates will be made after the state representations stabilize. Limited computational resources prevented our extended experiments, and we invite those with greater capacity to investigate further.

Our Dreamer implementations can be found in the source code repository.

