# OpenReview forum: "Rejecting Hallucinated State Targets during Planning"
_ICML.cc/2025/Conference — ICML 2025 poster_

### Official Review · Reviewer_bSxw · 2025-03-05

**Overall Recommendation:** 3

**Summary:**

This paper addresses the issue of hallucinated state targets in model-based reinforcement learning (MBRL), where generative models can produce unrealistic or unreachable states, leading agents to delusional planning behaviors. Inspired by human cognition, the authors propose an evaluator that assesses the feasibility of generated targets and rejects infeasible targets before planning occurs. To ensure accurate feasibility evaluation, the authors introduce two hindsight relabeling strategies, generat and pertask, demonstrating significant performance improvements and reduced delusional planning behaviors.

**Claims And Evidence:**

Yes

**Essential References Not Discussed:**

No

**Experimental Designs Or Analyses:**

I find the experimental designs reasonable and sufficiently aligned with the paper’s objectives.

**Methods And Evaluation Criteria:**

Yes

**Other Comments Or Suggestions:**

Some discussion on the computational overhead would be valuable.

**Other Strengths And Weaknesses:**

Strengths:

The paper addresses a critical issue in model-based RL—hallucinated targets—with implications for safety and performance.

The idea of combining a feasibility evaluator with hindsight relabeling strategies has intuitive appeal. Proposed approach is a simple, plugin mechanism that could help mitigate incorrect updates or infeasible sub-goal selection in a broad class of model-based agents.

The paper provides clear taxonomy of targets (G0, G1, G2) structures the problem effectively.

The paper includes multiple experiments—on both decision-time and background planning settings—demonstrating the reduction of delusional behaviors and improved performance.

Weaknesses:

While the grid-world-based tasks are carefully designed to showcase the phenomenon of hallucinations, they remain relatively simple compared to more complex, high-dimensional environments (e.g., robotics or rich 3D worlds). It is unclear if the proposed approach will scale without additional engineering. My major concern is that the experiments are mostly on tasks where there is a clear, easy-to-obtain ground-truth for feasibility. This is ideal for demonstrating concept correctness but may be less straightforward to replicate or validate in continuous or partially observable domains.

**Questions For Authors:**

How the feasibility evaluator performs in partially observable domains? Would the proposed approach naturally extend if the “source state” is uncertain?

If the generator produces more abstract or higher-dimensional latent goals (e.g., language instructions or subtask descriptors), what changes are required for the feasibility evaluator?"

**Relation To Broader Scientific Literature:**

The proposed approach builds upon prior model-based RL frameworks and opens the door to addressing the critical issue of hallucinated state targets through the explicit introduction of a feasibility evaluator trained via novel hindsight relabeling strategies.

**Theoretical Claims:**

No theoretical section is presented in the paper.

---

> ### Author Rebuttal · Authors · 2025-03-30
>
> `HOW EVALUATOR PERFORMS IN PARTIALLY OBSERVABLE DOMAINS? WOULD THE APPROACH NATURALLY EXTEND IF THE “SOURCE STATE” IS UNCERTAIN?`
>
> We agree with your intuition that it naturally extends.
>
> The evaluator takes on paired inputs of the source state representation and the target representation, both are outputs from the TAP agent to which the evaluator is attached. NO change is needed when the evaluator is fed representations (that can encode uncertainty) from a POMDP-compatible encoder.
>
> We are confident about this, yet we know it may be more convincing to conduct experiments to provide empirical evidence. Unfortunately, ICML prohibits revisions this time.
>
> `IF THE GENERATOR PRODUCES MORE ABSTRACT OR HD LATENT GOALS (E.G., LANGUAGE INSTRUCTIONS OR SUBTASK DESCRIPTORS), WHAT CHANGES ARE REQUIRED FOR THE EVALUATOR?`
>
> Since the evaluator takes in target representations, our design needs NO CHANGE. We validated this empirically in Sec. 5.3.
> We tried to make clear that our approach currently only applies to TAP agents with a function $h$ verifying if a target is fulfilled. TAP agents’ $h$ gives us the compatibility to general targets including language instructions.
>
> Instead of changes to the evaluator, the focus should be on a proper $h$ (e.g., ask the LM if a state matches the target). We discussed this at the end of Sec. 7.
>
> `THE GRID-WORLD-BASED TASKS REMAIN SIMPLE COMPARED TO MORE COMPLEX, HD ENVIRONMENTS. UNCLEAR IF THE APPROACH WILL SCALE WITHOUT ADDITIONAL ENGINEERING. … THIS … MAY BE LESS STRAIGHTFORWARD TO REPLICATE OR VALIDATE IN CONTINUOUS OR PARTIALLY OBSERVABLE DOMAINS.`
>
> Our reasons for the selected experiments are as follows:
> 1. These environments provide much more convincing quantitative validations of our claims on hallucinated targets and delusional estimates. Common benchmarks such as Atari, due to the lack of access to ground truth, cannot be properly diagnosed. As a result, we cannot prove that our method can indeed reduce delusional planning behaviors in those environments due to their nature.
> 2. We aimed to show our approach’s generality by applying it on many categories of TAP methods. The compute demanded in these experiments already exceeds what our limited academic environment provides.
> 3. Visual simplicity does not mean task simplicity. Due to the multi-task, generalization focused setting, agents are met with difficult combinatorial challenges that even state-of-the-art hierarchical planning methods cannot solve well, see [Zhao et al., 2024]. As an example, despite the visual simplicity, the hallucination rates remain high even with the used SOTA methods.
> For the points above, we focused on depth rather than breath when considering the environments for our experiments. Our approach does not assume the input space to be discrete or continuous. We chose discrete input spaces because it is nearly impossible to solve the ground truths, used to provide convincing, rigorous analytics, in continuous spaces.
>
> For POMDP, please check our prior responses.
>
> `DISCUSSIONS ON THE COMPUTATIONAL OVERHEAD WOULD BE VALUABLE.`
>
> Technically, it is useful to view our solution as a special form of rejection sampling, where the proposal distribution and the support are provided by the generator, the target distribution is the re-normalized distribution over only feasible targets and feasibility estimates by the evaluator is used to determine the rejection. Thus, the more accurate the evaluator, the more efficient the sampling process. It is impractical to assume full access to either the target or proposal distributions because they change with the used 1) environment and 2) generator (whose output is not only different per method but also changing with learning). This means that we cannot give a blanket statement about the computational overhead, prompting us to have used only brief discussions about overhead in Sec. 4 (L189, left col.).
>
> Practically, each generated target needs only 1 evaluation. For background TAP agents that generate batches of targets, the improper ones can be rejected and the whole batch can be all rejected without problem (no "Dyna" update this time). For decision-time TAP agents, targets act as subgoals and when they are rejected, the agent can retry or commit to more random explorations.
>
> With DRL, the overhead also depends on evaluator’s networks, complexity of the state / target representations. However, since the evaluator is a rather lightweight secondary end-to-end network, we can expect evaluations to be fast.
> We added a condensed version of these comments to the revised manuscript.
>
> ---
> We appreciate your review and your concerns about the applicability of our approach to more generic cases. We explained to you our choices and wish you could perceive the difficulties we overcame in acquiring the convincing results with limited resources.
>
> We hope our responses addressed your concerns well and please consider increasing your rating.

---

> > ### Comment · Reviewer_bSxw · 2025-04-05
> >
> > Thank you for the response and clarifications.
> >
> > I understand the reasoning behind the choice of grid-world tasks for rigorous analysis and quantitative validation, particularly regarding ground truth availability. That said, I still have concerns about how well proposed approach might generalize to more complex, high-dimensional environments without additional engineering. While the current experimental results are convincing for the selected tasks, it remains unclear how it would perform in other complex domains where ground-truth feasibility is not readily avaliable. Thus, I will maintain my current score (weak accept).

---

### Official Review · Reviewer_GvFL · 2025-03-13

**Overall Recommendation:** 2

**Summary:**

The paper proposes to augment Target-Assisted Planning (TAP) methods with an evaluator to reject  generated states that are unfeasible and improve performance. The proposed method is evaluated on two environments: SwordShieldMonster (SSM) and RandDistShift (RDS) with 3 different TAP agents: Dyna, Skipper and LEAP.

It proposes 4 different mechanisms to create targets for the evaluator.
Future (F), which sample states in the future
Episode (E), which sample states from the same episode
Generate (G), which replace states by their generated version predicted by the generator
Per task (P), which sample states from the same environment task

The experiments show that a combination of target mechanisms such as (E+P+G) helps to further reduce the evaluator prediction errors and improve final success rates on these 2 environments.

## update after rebuttal

I appreciate the authors clarification of the paper contributions. I understand that the algorithm tackles the rejection of hallucinated state predictions. The authors propose to learn an evaluator to reject these hallucinations. For that, the authors propose a combination of learning rules, including two novel relabelling strategies (PerTask and Generate). The technique is evaluated on two toy environments (SSM and RDS) providing game logic to prove the efficacy of the contributions.

As final decision:

I am still hesitant toward the generalization of the method. I think that the experiments done on the two environments are very compelling because they offer game logic for analysis. However, I think the paper would strongly benefit from experiments on commonly used benchmarks for model-based RL in order to provide general empirical results on top SSM and RDS (Comparison with concurrent approaches, ablation study on hallucination rejection). Visualizations of rejected predictions for some of the tasks would also help to highlight the method’s contribution.

I hence maintain my score to weak rejection, tending toward borderline.

**Claims And Evidence:**

The experiments show that the method can successfully reduce delusional behaviors and enhance the the performance of planning agents.

**Essential References Not Discussed:**

No

**Experimental Designs Or Analyses:**

Yes

**Methods And Evaluation Criteria:**

Yes, TAP and environments allows to correctly make into application the proposed method.

**Other Comments Or Suggestions:**

I tend toward borderline but would be ready to increase my score to weak accept or accept if additional experiments and/or information are provided.

**Other Strengths And Weaknesses:**

Strengths:
- The paper tackles a very interesting problem linked to TAP, which is the generation of infeasible states by world models. TAP methods usually suppose that all generated states are feasible and reduce the agent training loss on all generated samples. This research direction deserves to be explored.
- The paper demonstrates that learning an evaluator function can help to improve performance on the two chosen environments. The evaluator can successfully learn to progressively identify infeasible targets to remove them.

Weaknesses:
- The method is evaluated on only two simple environments (SSM and RDS). While the paper shows that performance can be improved by removing delusional predictions. It would be interesting to experiment on commonly used benchmarks such as Atari 100k or the DeepMind Control Suite.
- From my understanding, the labeling of targets as feasible or unfeasible requires having access to game inner logics (have sword, have shield). Such that we can correctly label sampled targets as feasible or unfeasible given the start state. I do not see how it could be applied to environments where game logic is not accessible.
- The paper talk about possible extensions to other TAP methods such as MuZero, SimPLe or Dreamer but do not perform experiments on these key methods.

**Questions For Authors:**

- Figure 3 (d), 10 (d) and 12 (d) compare the performance of Skipper and LEAP when using different labelling mechanisms. What is the performance of these methods when the evaluator is not used during training to remove predicted infeasible states ?

- From my understanding, the labeling of targets as feasible or unfeasible requires having access to game inner logics (have sword, have shield). Could this method be used on environments where game logics is not available to label states as feasible/infeasible ?

**Relation To Broader Scientific Literature:**

The contribution of this paper is related to model-based reinforcement learning agents that may suffer from hallucinations during the generation process. Early training of model-based approaches can sometimes impacted by the lower generation quality of the world model.

**Theoretical Claims:**

Yes

---

> ### Author Rebuttal · Authors · 2025-03-30
>
> `WHAT IS THE PERFORMANCE OF SKIPPER & LEAP  WHEN THE EVALUATOR IS NOT USED TO REMOVE PREDICTED INFEASIBLE STATES?`
>
> As discussed in Sec. 2 (Line 57 right column), Sec 5.1.2 (L383 left col.) , Sec. 6 (L388 r. col.), these methods already use their own evaluators to remove the targets they think that are infeasible. Yet, they suffer from delusions due to the lack of exposure to truly infeasible targets absent from collected experience.
>
> Thus, our exps with both methods "reused" their built-in distance / discount estimators, which were *originally trained with basic relabeling strategies*. We showed that by fixing their relabeling, these methods become robust against delusional planning behaviors. Only for methods without built-in estimators at all, e.g., Dyna, we have baselines with the evaluator removed.
>
> In other words, exps with Skipper & LEAP focuses on the importance of training data for non-delusional estimates. While exps 5/8 - 8/8 validate the combination of architecture, learning rules and training data at the same time.
>
> Hope that you can now see that your wanted baselines are already there: they are the variants with "future" and "episode" (basic relabeling that causes delusions).
>
> We realized that this could be somewhat confusing and tried to clarify in the revised manuscript.
>
> `THE METHOD IS EVALUATED ON ONLY 2 SIMPLE ENVIRONMENTS (SSM AND RDS). WHILE THE PAPER SHOWS THAT PERFORMANCE CAN BE IMPROVED BY REMOVING DELUSIONAL PREDICTIONS. IT WOULD BE INTERESTING TO EXPERIMENT ON COMMONLY USED BENCHMARKS …`
>
> Our reasons for the selected experiments are as follows:
> 1. These environments provide much more convincing quantitative validations of our claims on hallucinated targets and delusional estimates. Common benchmarks such as Atari, due to the lack of access to ground truth, cannot be properly diagnosed. As a result, we cannot prove that our method can indeed reduce delusional planning behaviors in those environments due to their nature.
> 2. We aimed to show our approach’s generality by applying it on many categories of TAP methods. The compute demanded in these experiments already exceeds what our limited academic environment provides.
> 3. Visual simplicity does not mean task simplicity. Due to the multi-task, generalization focused setting, agents are met with difficult combinatorial challenges that even state-of-the-art hierarchical planning methods cannot solve well, see [Zhao et al., 2024]. As an example, despite the visual simplicity, the hallucination rates remain high even with the used SOTA methods.
> For the points above, we focused on depth rather than breath when considering the environments for our experiments.
>
>
> `FROM MY UNDERSTANDING, THE LABELING OF TARGETS AS FEASIBLE OR UNFEASIBLE REQUIRES HAVING ACCESS TO GAME INNER LOGICS (HAVE SWORD, HAVE SHIELD). SUCH THAT WE CAN CORRECTLY LABEL SAMPLED TARGETS AS FEASIBLE OR UNFEASIBLE GIVEN THE START STATE. I DO NOT SEE HOW IT COULD BE APPLIED TO ENVIRONMENTS WHERE GAME LOGIC IS NOT ACCESSIBLE.`
>
> We respectfully note that this is a misunderstanding; our approach distinguishes itself from methods that require access to more environmental mechanics. The proposed evaluator figures out the feasibility of all targets through Eq. (2) without the need for labels: **Eq. (2) exploits the fact that infeasible targets will never be reached**.
>
> Eq. (2) enables the evaluator to learn as a secondary system alongside the TAP agent to which it is attached, and figure out from data, the feasibility of the sampled targets it is trained on.
>
> Ground truths are only used for experimental analyses, our solution operates without ANY.
>
> `THE PAPER TALK ABOUT POSSIBLE EXTENSIONS TO OTHER TAP METHODS SUCH AS MUZERO, SIMPLE OR DREAMER BUT DO NOT PERFORM EXPERIMENTS ON THESE KEY METHODS.`
>
> In Table 2, we categorized the compatible methods into 3 categories of similar behaviors, and in experiments, **we implemented one representative method for each of the categories**.
>
> We made a conscious effort to provide you with the most convincing results out of the limited resources at our disposal. We hope that you understand our difficulties in an academic setting.
>
> ---
> Thank you for your comments. Your biggest concern was about the applicability of the method without access to feasibility labeling. We hope that our explanations will address our miscommunication on this point. For your comments on experiments, we clarified that the certain baselines in the mentioned figures are the original performance of the agents, as both Skipper and LEAP are equipped with built-in feasibility-like estimators, which we dual-purposed as evaluators.
>
> We are thankful that you acknowledged that “this direction deserves to be explored” and hope that our responses have addressed your concerns, and you could increase your rating of this work to recognize our contributions.

---

> > ### Comment · Reviewer_GvFL · 2025-04-05
> >
> > Thank you for your response.
> >
> > Your rebuttal addressed my misunderstanding related to the contributions of the paper, choice of environment used and comparisons.
> >
> > It appears that the main contributions of the paper are (not to identify hallucinated model predictions, which was already proposed by Nasiriany et al. (2019); Zhao et al. (2024); Lo et al. (2024)), but:
> > - 1. an evaluator model that do not require ground truths for training, in contrast to previous approaches
> > - 2. the combination of labeling strategies to decrease feasibility error and improve performance
> > - 3. two novel labeling strategies (PerTask and generate)
> >
> > This somewhat downgrades the estimated novelty of the paper compared to my initial review. Moreover, reviewer m9Mj pointed out that "Generate" and "PerTask" are similar to previous works (Zawalski et al., Andrychowicz et al.), which downgrades contribution number 3.
> >
> > In light of these clarifications on the paper contributions, I see the introduction of a hallucination detection method that does not require "ground truths" as notably more insightful for the community than the contributions highlighted in the paper. This reinforces the initial point that I made on the application of the method on Atari 100k. I think the paper would greatly benefit from the application of the method on commonly used environments that do not provide ground truths / inner logic.  Proving the effectiveness of rejecting hallucinations on diverse and commonly acknowledged benchmarks like Atari 100k, Crafter on top of the existing SSM and RDS experiments would provide general empirical results on the effectiveness of the method rather than a proof a concept.
> >
> > I choose to maintain my initial weak reject rating leaning more toward borderline, but I do not oppose the paper to be accepted.
> >
> > W1: I think the paper proposes an interesting solution to reject hallucination without access to environment inner logics but instead highlights orthogonal novelties that are less notable.
> >
> > W2: The benchmarking of the method is limited due to the comparison with previous approaches that requires "ground truths", also due to the lack of computing resources.

---

> > > ### Author Response · Authors · 2025-04-05
> > >
> > > We really appreciate your comments and your open-mindedness to our explanations.
> > >
> > > First, we sincerely agree to your points on more general empirical results. But as we explained, we are shy of resources to make it happen. **We'd like to point out that several claims in your newest response are not factual and we believe, based on the good will you have shown in your reply, our explanations here would make you find our contribution to be more than acceptance-deserving**.
> > >
> > > ---
> > >
> > > `not to identify hallucinated model predictions, which was already ...`
> > >
> > > The 3 works mentioned here did NOT identify hallucinated model predictions. In fact, **this submission is indeed the 1st work that systematically studies hallucinated targets in planning**. Our previous explanations (on Skipper & LEAP having their built-in estimators to identify infeasible targets) may have confused you. As investigated in this work, there are different types of infeasible targets, notably including 1) those that appear in the interaction history (which Skipper and LEAP could identify) and **2) those that are never going to be experienced by agents** (hallucinations, that most existing methods, including the two, CANNOT identify). The latter kind is the focus of this work.
> > >
> > > `1. an evaluator model that do not require ground truths for training, in contrast to previous approaches / W2: ... due to the comparison with previous approaches that require ground truths`
> > >
> > > We would like to be honest and point out that the learning rules in existing approaches, e.g., Skipper, already do not assume access to ground truths (to figure out the feasibility of targets). Yet, what distinguishes this work from the previous ones is that **despite the proper auto-discovery learning rules, previous methods will not lead to correct understanding of the hallucinated targets that are never going to be experienced** (the latter type in the previous point). This work identified how such feasibility delusion (*delusions are errors persistent due to design flaws and cannot be addressed by more training*. We use these terms rigorously) is formed and proposed to use the relabeling strategies to provide correct data exposure.
> > >
> > > W2: The previous approaches do NOT need the access to ground truths, as they are not even aware of the hallucinated targets. Rather, we used these ground truths to obtain quantitative metrics to show that 1) hallucinated targets exist, 2) they cause delusions, which causes delusional plans 3) addressing them leads to better performance in many TAP methods. We believe that proving our approach's effectiveness directly is more convincing than blindly showing only the performance boosts. On Atari, this would've been impossible.
> > >
> > > `2. ... to decrease feasibility error and improve performance`
> > >
> > > As explained previously, our contribution addresses the feasibility delusions (the portion of feasibility error corresponding to the hallucinated targets, un-learnable by most existing approaches).
> > >
> > > `strategies are similar to ... existing ... downgrades contribution`
> > >
> > > The premise of this work is to raise awareness of the problem of hallucinated targets within a wide range of methods (TAP). We want to provide you with our thoughts on the contributions of this work:
> > >
> > > - As a first, we systematically investigated the properties and impact of different kinds of infeasible targets. Most notably, G1 and G2 and the generic set correspondence. Guided by analyses, we devised a generic target evaluator that rejects infeasible targets (both kinds) that can work for many TAP agents with different planning behaviors.
> > >
> > > - We shared our desiderata, in that the evaluator should act as an add-on **without the need to change the behavior nor the architectures of the agent it is attached to**.
> > >
> > > - We highlighted that, without proper training, the evaluator WILL produce **delusional estimates**, just in existing methods such as Skipper, rendering the evaluator-based solution futile. Notably, many TAP methods such as those with fixed planning horizons do not have a feasibility-like estimator to begin with and blindly accepts all generated targets.
> > >
> > > - From the data exposure perspective, we analyzed why learned evaluators become delusional. And we proposed to use 1) 2 alternative relabeling strategies that work hand-in-hand with 2) an efficient architecture with distributional outputs and 3) off-policy compatible learning rules capable of discovering the feasibility of **all exposed targets** (most existing methods are oblivious to the latter kind of hallucinated targets).
> > >
> > > - Our experiments validate significant reductions in delusional behaviors and enhancements in the performance of several kinds of TAP agents.
> > >
> > > ---
> > >
> > > Your reply gave us a glimmer of hope that this work may be accepted, which we firmly believe that it deserves. **We believe that your current evaluation is still impacted by some miscommunications, and that is why we are taking this urgent reply for you to reconsider.** Thank you!

---

### Official Review · Reviewer_m9Mj · 2025-03-14

**Overall Recommendation:** 3

**Summary:**

The paper analyzes the issue of generating invalid subgoals during planning. The authors categorize different failure modes and propose strategies for learning a classifier that can be used to estimate the distance to a proposed goal, including whether it is reachable at all. Through experimental evaluation in a grid-based task, the paper analyzes the impact of different learning strategies on the effectiveness of the evaluator.

## Update after rebuttal

Thank you for the answers. I acknowledge the differences between the proposed strategies and those present in the literature. However, I believe these differences need to be discussed more precisely in the paper to better highlight the contribution -- as noted by other reviewers as well, this is not currently clear. I leave the particular choice of references to the authors' choice.

I acknowledge the authors' focus on evaluating the benefits of a non-delusional evaluator compared to a standard one. However, completely omitting the non-evaluator aspect makes the analysis incomplete. Even if performance without the evaluator is significantly weaker, this should be demonstrated and briefly remarked. There is value in advocating for a non-delusional evaluator only if using an evaluator is beneficial in the first place, even if the overall focus of the paper is slightly different.

In summary, I believe the paper is a solid contribution, but a careful revision would considerably strengthen its impact. Specifically, I suggest:

- Revising the stated contributions as discussed,

- Clarifying the novelty of the proposed strategies,

- Including a naive non-evaluator baseline in the comparison,

- Adding experiments in widely studied environments.

With such improvements, I would consider the paper very strong. For now, I remain on the fence.

I acknowledge the changes proposed by the authors, which seem to move in the right direction. I reflect this by increasing my rating. I would not oppose accepting the paper, though I believe it could still be significantly strengthened with minor effort.

**Claims And Evidence:**

The paper is generally sound, although the main claims should be reformulated, as they are too optimistic. The idea of training a model to identify infeasible subgoals was already proposed, although possibly not extensively studied (see e.g. [Zawalski et al.]). Furthermore, the proposed relabeling strategies are not "novel", as Generate is similar to the one used e.g. in [Zawalski et al.], and Pertask is equivalent to Random from [Andrychowicz et al.]. While itself it does not deny the contribution of the paper, the formulation of the main contributions has to be revised. I believe that is why the paper lacks focus.

Additionally, the paper only indirectly argues that identifying the invalid subgoals is useful in general. For instance, the main claims do not state any performance advantage, and the experiments focus on comparing different strategies of training the evaluator rather then the impact of evaluator on performance. The only somewhat relevant plot is Fig 3d (and counterparts in the appendix). A very good step in this direction is Section 5.2. The paper would strongly benefit from including more evaluation of this kind, i.e. demonstrating that various methods can do much better having the evaluator. Currently I see little such discussion, which is a pity.

References:

[Zawalski et al.] _Fast and Precise: Adjusting Planning Horizon with Adaptive Subgoal Search_

[Andrychowicz et al.] _Hindsight Experience Replay_

**Essential References Not Discussed:**

One additional reference I would like to be discussed is [Zawalski et al.], as detailed in the comments.

**Experimental Designs Or Analyses:**

Yes, I checked the experiments in the main part.

**Methods And Evaluation Criteria:**

The presented evaluation makes sense for the formulated main claims. However, slightly changing the scope to cover the usefullness of the evaluator, possibly also in well-established benchmarks, would improve the contribution.

**Other Comments Or Suggestions:**

Are the targets for the Generate strategy additionally generated during sampling from buffer, or are they the targets generated by the generator during collecting the experience and stored afterwards? I suppose the former, but that should be made clear in Sec 4.2.1.

**Other Strengths And Weaknesses:**

While the paper has potential, it should be much more focused. Too little space is reserved for experiments, and because of that many of them had to be moved to appendix. I suggest making the analysis more concise and providing broader experimental support. I suggest working on the main contributions to establish the focus. Something around ["Systematic analysis of infeasible subgoals issue", "Effective training and architecture for Evaluator", "Experimentally validating the impact of Evaluator on performance"] could be a good starting point.

**Questions For Authors:**

1. Please discuss the impact of having the evaluator on the performance of hierarchical methods. The naive approach to invalid targets is to ignore them, as they are not very common, and even invalid subgoal guidance lead somewhere, from where another (hopefully valid) sobgoal can be generated. Are the methods equipped with Evaluator more effective? What is its computational overhead?
2. Please discuss the relation of the proposed relabeling strategies (claimed to be novel) to [Zawalski et al.] and [Andrychowicz et al.]

**Relation To Broader Scientific Literature:**

The problem of detecting invalid subgoals is not new and has been (at least partially) studied in previous works. Variants of the proposed strategies for learning the evaluator can be also found in related works. However, I am not aware of a systematic study of this topic, so it has the potential to be a good contribution.

**Theoretical Claims:**

There is one theorem: Result 4.1. No proof is referenced, but since it is rather straightforward, it needs no proof.

---

> ### Author Rebuttal · Authors · 2025-03-30
>
> We reordered the questions to streamline our response.
>
> `Discuss the relation … to [Z et al.] & [A et al.]. Are the targets for “generate” generated during sampling, or …?`
>
> [Z et al.]
>
> Our approach of rejecting infeasible targets is indeed similar to that in [Z et al.]. Yet, the approaches differ significantly. [Z et al.]’s verifier trains on a collected dataset that informs about the generated subgoals (not relabeled) & requires separate training for the verifier.
> Unlike the method-specific approach in [Z et al.], we propose a generally applicable secondary system running alongside TAP agents, using relabeling, and thus it can figure out the infeasible targets (sets of states) that were not followed, without extra interactions or supervised training.
>
> We wanted to try our best to acknowledge the developments leading to ours. This is why we discussed MHER (Yang et al., 2021) in Sec 6 for the 1st work of model-based to relabeling transitions (even closer to “generate”) and (Jaferjee et al., 2020) for their early idea of rejecting delusional generations. It was an *honest mistake* that we missed [Z et al.] and we now added proper acknowledgements.
>
> "generate"& JIT relabeling
>
> "generate" is novel not only for its flexibility to be used just-in-time (JIT, only relabeling after batch is sampled to adhere to the generators’ changing output distributions), but also for its effectiveness to address feasibility delusions. Compared to approaches like MHER, "generate" lowers needed storage and provides timely coverage of the generators’ outputs, especially helpful in continual learning settings.
>
> "pertask" & [A et al.]
>
> "pertask" is ONLY equivalent to “random” in [A et al.] when agents are trained on a single task. In settings where agents are trained on a few tasks and are expected to generalize during evaluation (where our exps were based, to force the evaluators to understand infeasible targets, instead of memorize), "pertask" enables relabeling beyond trajectory-level against delusions, per Sec 5.1.
>
> `the exps focus on strategies of training rather than the impact of evaluator on performance. … only indirectly argues that identifying invalid subgoals is useful. the main claims do not state any performance advantage … the impact of the evaluator on the performance of hierarchical methods?`
>
> **This is a crucial misunderstanding that we wish to clarify, so we can resolve your other concerns effectively.**
>
> Our aim was not to show methods can do better with an evaluator, but a non-delusional one.
> Exps on the two HP agents focused on the fact that their basic relabeling strategies produce delusional evaluators (they have built-in evaluators). Only for methods without estimators to begin with, e.g. Dyna, we need to completely inject an evaluator. We discussed these explicitly in Sec 2, Sec 5.1.2 & Sec 6.
>
> Due to the reply limit, plz find MORE details on this, in the 1st reply to Reviewer GvFL.
>
> We added clarifications in the revision.
>
> `The naive approach to invalid targets is to ignore, as they aren’t very common, and even invalid subgoal guidance lead somewhere, from where another subgoal can be generated.`
>
> In Sec 2, we formulated your described scenario for the naïve approach, which motivated our contributions. The naïve approach only works when 1) infeasible targets are indeed rare and 2) the state space is simple. Also, the approach is boosted by 3) survivorship bias.
> 1. There are generally no measures of how frequent infeasible targets can appear, because most existing methods are not tested on proper environments, that can be solved to produce the true frequency of infeasible targets.
> 2. it is not hopeful that invalid guidance could lead agents to a recoverable region of a complex state space
> 3. Empirical observations of how naïve approaches could work are influenced by survivorship bias, as methods that are more affected by delusional plans have worse performance and are thus less likely to be recognized.
>
> `computational overhead?`
>
> Plz see our reply to Reviewer bSxw (last point).
>
> `the main claims should be reformulated, as they are too optimistic. The idea … to identify infeasible subgoals was already proposed … `
>
> We hope our previous explanations addressed your concerns on this point.
> We summarized our work’s claims and tried our best to clarify in the revision:
>
> Building on the ideas of rejecting invalid goals such as in [Z et al.], our extensive study reveals the types of infeasible targets in TAP agents. Accordingly, the generality of our proposed solution leads to non-delusional feasibility estimates beyond HP methods.
>
> ---
> We appreciate your detailed review!
> The intention of this work is to inform the research community about the associated risks and improper assumptions and save everyone’s time and effort.
> We take your comments very seriously and tried our best to address your concerns. We hope you can increase your rating to recognize our attitude and the positive impact this work could have.

---

### Decision · Program_Chairs · 2025-05-01

**Decision:**

Accept (poster)

**Comment:**

The paper presents a systematic analysis of the properties and impact of different types of infeasible targets in Target-Assisted Planning (TAP) agents. The authors introduce a generic target evaluator capable of rejecting targets across a range of TAP agents. Importantly, the evaluator is designed as an add-on module that requires no modifications to the agent’s behavior or architecture, but it does require access to the same experiences. The authors analyzed the causes of evaluator delusions and proposed several methods to assess feasibility across exposed targets. Experimental results demonstrate significant reductions in delusional behavior and improved performance across various TAP agents.

This paper was reviewed by three reviewers, with broad agreement that the claims and contributions are significant and well-supported by extensive experimental evaluation.

The author response phase was intensive and very helpful in clarifying the key contributions and novelty of the work. I acknowledge the efforts by the authors in making these aspects clearer during the rebuttal. All reviewers agreed that a revised version should clarify the contributions and novelty. This was addressed in the revised version I considered for writing the meta-review.

However, all reviewers shared a concern, and I agree with them, about the limited generality of the evaluated tasks, which focus exclusively on grid-like domains. While the reasons outlined by the authors for considering this environment are appropriate and convincing for evaluating their methodology, it remains unclear whether the proposed methodology and associated computational overhead are necessary or feasible beyond these domains, given that the main purpose is simply to reject hallucinated states (as the authors state in Fig. 1 and Section 4.3 of the revised manuscript).

These considerations reduce the impact of their work, and based on them, I recommend a weak accept.